# OSCILLATORY STATE-SPACE MODELS

**T. Konstantin Rusch**
MIT
tkrusch@mit.edu

**Daniela Rus**
MIT

## ABSTRACT

We propose Linear Oscillatory State-Space models (LinOSS) for efficiently learning on long sequences. Inspired by cortical dynamics of biological neural networks, we base our proposed LinOSS model on a system of forced harmonic oscillators. A stable discretization, integrated over time using fast associative parallel scans, yields the proposed state-space model. We prove that LinOSS produces stable dynamics only requiring nonnegative diagonal state matrix. This is in stark contrast to many previous state-space models relying heavily on restrictive parameterizations. Moreover, we rigorously show that LinOSS is universal, i.e., it can approximate any continuous and causal operator mapping between time-varying functions, to desired accuracy. In addition, we show that an implicit-explicit discretization of LinOSS perfectly conserves the symmetry of time reversibility of the underlying dynamics. Together, these properties enable efficient modeling of long-range interactions, while ensuring stable and accurate long-horizon forecasting. Finally, our empirical results, spanning a wide range of time-series tasks from mid-range to very long-range classification and regression, as well as long-horizon forecasting, demonstrate that our proposed LinOSS model consistently outperforms state-of-the-art sequence models. Notably, LinOSS outperforms Mamba and LRU by nearly 2x on a sequence modeling task with sequences of length 50k. Code: https://github.com/tk-rusch/linoss.

## 1 INTRODUCTION

State-space models (Gu et al., 2021; Hasani et al., 2022; Smith et al., 2023; Orvieto et al., 2023) have recently emerged as a powerful tool for learning on long sequences. These models posses the statefullness and fast inference capabilities of Recurrent Neural Networks (RNNs) together with many of the benefits of Transformers (Vaswani, 2017; Devlin, 2018), such as efficient training and competitive performance on large-scale language and image modeling tasks. For these reasons, state-space models have been successfully implemented as foundation models, surpassing Transformer-based counterparts in several key modalities, including language, audio, and genomics (Gu & Dao, 2023).

Originally, state-space models have been introduced to modern sequence modelling by leveraging specific structures of the state matrix, i.e., normal plus low-rank HiPPO matrices (Gu et al., 2020; 2021), allowing to solve linear recurrences via a Fast Fourier Transform (FFT). This has since been simplified to only requiring diagonal state matrices (Gu et al., 2022a; Smith et al., 2023; Orvieto et al., 2023) while still obtaining similar or even better performance. However, due to the linear nature of state-space models, the corresponding state matrices need to fulfill specific structural properties in order to learn long-range interactions and produce stable predictions. Consequently, these structural requirements heavily constrain the underlying latent feature space, potentially impairing the model's expressive power.

In this article, we adopt a radically different approach by observing that forced harmonic oscillators, the basis of many systems in physics, biology, and engineering, can produce stable dynamics while at the same time seem to ensure expressive representations. Motivated by this, we propose to construct state-space models based on stable discretizations of forced linear second-order ordinary differential equations (ODEs) modelling oscillators. Our additional contributions are:

- we introduce implicit and implicit-explicit associative parallel scans ensuring fast training and inference.

- we show that our proposed state-space model yields stable dynamics and is able to learn long-range interactions only requiring nonnegative diagonal state matrix.

- we demonstrate that a symplectic discretization of our underlying oscillatory system conserves its symmetry of time reversibility.

- we rigorously prove that our proposed state-space model is a universal approximator of continuous and causal operators between time-series.

- we provide an extensive empirical evaluation of our model on a wide variety of sequential data sets with sequence lengths reaching up to 50k. Our results demonstrate that our model consistently outperforms or matches the performance of state-of-the-art state-space models, including Mamba, S4, S5, and LRU.

## 2  THE PROPOSED STATE-SPACE MODEL

Our proposed state-space model is based on the following system of forced linear second-order ODEs together with a linear readout,

$$\begin{aligned} \mathbf{y}''(t) &= -\mathbf{A}\mathbf{y}(t) + \mathbf{B}\mathbf{u}(t) + \mathbf{b}, \\ \mathbf{x}(t) &= \mathbf{C}\mathbf{y}(t) + \mathbf{D}\mathbf{u}(t), \end{aligned} \tag{1}$$

with hidden state $\mathbf{y}(t) \in \mathbb{R}^m$, output state $\mathbf{x}(t) \in \mathbb{R}^q$, time-dependent input signal $\mathbf{u}(t) \in \mathbb{R}^p$, weights $\mathbf{A} \in \mathbb{R}^{m \times m}$, $\mathbf{B} \in \mathbb{R}^{m \times p}$, $\mathbf{C} \in \mathbb{R}^{q \times m}$, $\mathbf{D} \in \mathbb{R}^{q \times p}$, and bias $\mathbf{b} \in \mathbb{R}^m$. Note that $\mathbf{A}$ is a diagonal matrix, i.e., with non-zero entries only on its diagonal. We further introduce an auxiliary state $\mathbf{z}(t) \in \mathbb{R}^m$, with $\mathbf{z} = \mathbf{y}'$. We can thus write (1) (omitting the bias $\mathbf{b}$ and linear readout $\mathbf{x}$) equivalently as,

$$\begin{aligned} \mathbf{z}'(t) &= -\mathbf{A}\mathbf{y}(t) + \mathbf{B}\mathbf{u}(t), \\ \mathbf{y}'(t) &= \mathbf{z}(t). \end{aligned} \tag{2}$$

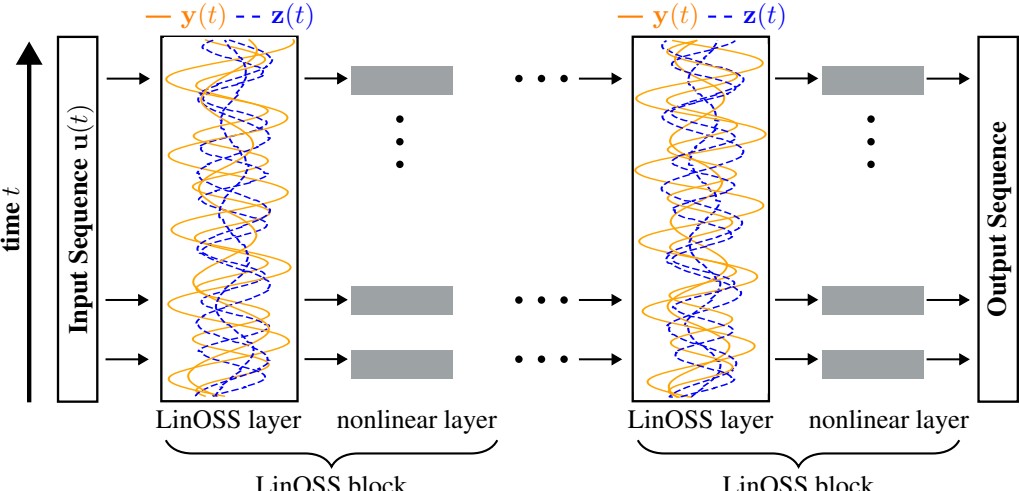

Figure 1: Schematic drawing of the proposed Linear Oscillatory State-Space model (LinOSS). The input sequences are processed through multiple LinOSS blocks. Each block is composed of a LinOSS layer (2) ($\mathbf{y}(t)$ plotted using orange solid lines and $\mathbf{z}(t)$ using blue dashed lines) followed by a nonlinear transformation, specifically a Gated Linear Units (Dauphin et al., 2017) (GLU) layer in our case. After passing through several LinOSS blocks, the latent sequences are decoded to produce the final output sequence.

### 2.1  MOTIVATION AND BACKGROUND

To demonstrate that the underlying ODE in (1) models a network of forced harmonic oscillators, we begin with the scalar case by setting $p = m = 1$ in (1). Choosing $\mathbf{B} = \mathbf{b} = 0$, we obtain the classic

ODE, $\mathbf{y}'' + \mathbf{A}\mathbf{y} = 0$, which describes simple harmonic motion with frequency $\mathbf{A} > 0$, such as that of a simple pendulum (Guckenheimer & Holmes, 1990). Now, allowing $\mathbf{B} \neq 0$ introduces external forcing proportional to the input signal $\mathbf{u}(t)$, where $\mathbf{B}$ modulates the effect of the forcing. Finally, setting $p, m > 1$ yields an uncoupled system of forced harmonic oscillators.

**Neuroscience inspiration.** Our approach is inspired by neurobiology, where the periodic spiking and firing of action potentials in individual neurons can be observed and analyzed as oscillatory phenomena. Furthermore, entire networks of cortical neurons exhibit behavior-dependent oscillations, reviewed in Buzsaki & Draguhn (2004). Remarkably, despite their complexity, these neural oscillations share characteristics with harmonic oscillators, as described in Winfree (1980). Building on this insight, we distill the core essence of cortical dynamics and aim to construct a machine learning model following the motion of harmonic oscillations. Finally, we note that our focus is on the theoretical and empirical aspects of the proposed oscillatory state-space model in this manuscript. However, our model can be further used in the context of computational neuroscience emulating characteristic phenomena of brain oscillations, such as frequency-varying oscillations, transient synchronization or desynchronization of discharges, entrainment, phase shifts, and resonance, while at the same time being able to learn non-trivial input-output relations.

## 2.2 COMPARISON WITH RELATED WORK

State-space models have seen continuous advancements since their introduction to modern sequence modeling in Gu et al. (2021). The original S4 model (Gu et al., 2021), along with its adaptations (Gu et al., 2022a; Nguyen et al., 2022; Goel et al., 2022), utilized FFT to solve linear recurrences. More recently, a simplified variant, S5 (Smith et al., 2023), was introduced, which instead employs associative parallel scans to achieve similar computational speed. Another advancement to S4 have been Liquid Structural State-Space models (Hasani et al., 2022) which exchanged the static state matrix with an input-dependent state transition module. While all aforementioned models rely on the HiPPO parameterization (Gu et al., 2020), Linear Recurrent Units (LURs) (Orvieto et al., 2023) demonstrated that even simpler parameterization yield state-of-the-art performance. Finally, selective state spaces have been introduced in Gu & Dao (2023) in order to further close the performance gap between state-space models and Transformers utilized in foundation models.

Oscillatory dynamics as a neural computational paradigm has been originally introduced via Coupled Oscillatory RNNs (CoRNNs) (Rusch & Mishra, 2021a). In this work, it has been shown that recurrent models based on nonlinear oscillatory dynamics are able to learn long-range interactions by mitigating the exploding and vanishing gradients problem (Pascanu, 2013). This approach was later refined for handling very long sequences in Rusch & Mishra (2021b), which utilized uncoupled nonlinear oscillators for fast sequence modeling by adopting a diagonal hidden-to-hidden weight matrix. Interestingly, this method resembles modern state-space models but distinguishes itself by employing nonlinear dynamics. Since then, this concept, generally termed as neural oscillators (Lanthaler et al., 2024), has been extended to other areas of machine learning, e.g., Graph-Coupled Oscillator Networks (GraphCON) (Rusch et al., 2022) for learning on graph-structured data, and RNNs with oscillatory dynamics combined with convolutions leading to locally coupled oscillatory RNNs (Keller & Welling, 2023). Our proposed LinOSS model differs from all these methods by its explicit use of harmonic oscillators via a state-space model approach.

## 2.3 DISCRETIZATION

Our aim is to approximately solve the linear ODE system (2) as fast as possible, while at the same time being able to guarantee stability over long time-scales. To this end, we suggest to leverage the following two time integration schemes.

**Implicit time integration.** We fix a timestep $0 < \Delta t \leq 1$ and define our proposed state-space model hidden states at time $t_n = n\Delta t$ as the following implicit discretization of the first order system (2):

$$\mathbf{z}_n = \mathbf{z}_{n-1} + \Delta t(-\mathbf{A}\mathbf{y}_n + \mathbf{B}\mathbf{u}_n),$$
$$\mathbf{y}_n = \mathbf{y}_{n-1} + \Delta t\mathbf{z}_n.$$

This can be written in matrix form by introducing $\mathbf{x}_n = [\mathbf{z}_n, \mathbf{y}_n]^\top$,

$$\mathbf{M}\mathbf{x}_n = \mathbf{x}_{n-1} + \mathbf{F}_n,$$

with

$$\mathbf{M} = \begin{bmatrix} \mathrm{I} & \Delta t \mathbf{A} \\ -\Delta t \mathrm{I} & \mathrm{I} \end{bmatrix}, \qquad \mathbf{F}_n = \begin{bmatrix} \Delta t \mathbf{B} \mathbf{u}_n \\ \mathbf{0} \end{bmatrix}.$$

We can now solve our discretized oscillatory system by simply inverting matrix $\mathbf{M}$ and computing the induced recurrence. Note that inverting matrices typically requires $\mathcal{O}(m^3)$ operations (where $m$ is the number of rows or columns of the matrix) using methods like Gauss-Jordan elimination, making it computationally expensive. However, by leveraging the Schur complement, we can obtain an explicit form of $\mathbf{M}^{-1}$ that can be computed in $\mathcal{O}(m)$, thanks to the diagonal structure of $\mathbf{A}$. More concretely,

$$\mathbf{M}^{-1} = \begin{bmatrix} \mathrm{I} - \Delta t^2 \mathbf{A}\mathbf{S} & -\Delta t \mathbf{A}\mathbf{S} \\ \Delta t \mathbf{S} & \mathbf{S} \end{bmatrix} = \begin{bmatrix} \mathbf{S} & -\Delta t \mathbf{A}\mathbf{S} \\ \Delta t \mathbf{S} & \mathbf{S} \end{bmatrix}, \tag{3}$$

with the inverse of the Schur complement $\mathbf{S} = (\mathrm{I} + \Delta t^2 \mathbf{A})^{-1}$ which itself is a diagonal matrix and can thus be trivially inverted. We point out that among other choices, a straightforward condition that ensures $\mathbf{S}$ is well-defined is $\mathbf{A}_k \geq 0$ for all $k = 1, \ldots, m$. The recurrence of the proposed model is then given as,

$$\mathbf{x}_n = \mathbf{M}^{\mathrm{IM}}\mathbf{x}_{n-1} + \mathbf{F}_n^{\mathrm{IM}}, \tag{4}$$

with $\mathbf{M}_n^{\mathrm{IM}} = \mathbf{M}^{-1}$, and $\mathbf{F}_n^{\mathrm{IM}} = \mathbf{M}^{-1}\mathbf{F}_n$. As we will show in the subsequent section, this discretization leads to a globally asymptotically stable discrete dynamical system.

**Implicit-explicit time integration (IMEX).** Another discretization yielding stable dynamics that, however, do not converge exponentially fast to a steady-state can be obtained by leveraging symplectic integrators. To this end, we fix again a timestep $0 < \Delta t \leq 1$ and define our proposed state-space model hidden states at time $t_n = n\Delta t$ as the following implicit-explicit (IMEX) discretization of the first order system (2):

$$\begin{aligned} \mathbf{z}_n &= \mathbf{z}_{n-1} + \Delta t(-\mathbf{A}\mathbf{y}_{n-1} + \mathbf{B}\mathbf{u}_n), \\ \mathbf{y}_n &= \mathbf{y}_{n-1} + \Delta t \mathbf{z}_n. \end{aligned} \tag{5}$$

The only difference compared to the previous fully implicit discretization is the explicit treatment of the hidden state $\mathbf{y}$ (i.e., using $\mathbf{y}_{n-1}$ instead of $\mathbf{y}_n$) in the first equation of (5). As before, we can simplify this system in matrix form,

$$\mathbf{x}_n = \mathbf{M}^{\mathrm{IMEX}}\mathbf{x}_{n-1} + \mathbf{F}_n^{\mathrm{IMEX}}, \tag{6}$$

with

$$\mathbf{M}^{\mathrm{IMEX}} = \begin{bmatrix} \mathrm{I} & -\Delta t \mathbf{A} \\ \Delta t \mathrm{I} & \mathrm{I} - \Delta t^2 \mathbf{A} \end{bmatrix}, \qquad \mathbf{F}_n^{\mathrm{IMEX}} = \begin{bmatrix} \Delta t \mathbf{B} \mathbf{u}_n \\ \Delta t^2 \mathbf{B} \mathbf{u}_n \end{bmatrix}.$$

Interestingly, ODE system (2) represents a Hamiltonian system (Arnold, 1989), with Hamiltonian,

$$\mathbf{H}(\mathbf{y}, \mathbf{z}, t) = \frac{1}{2} \sum_{k=1}^{m} \mathbf{A}_k \mathbf{y}_k^2 + \mathbf{z}_k^2 - 2 \left( \sum_{l=1}^{p} \mathbf{B}_{kl} \mathbf{u}(t)_l \right) \mathbf{y}_k. \tag{7}$$

The numerical approximation of this system using the previously described IMEX discretization is symplectic, i.e., it preserves a Hamiltonian close to the Hamiltonian of the continuous system. Thus, by the well-known Liouville's theorem (Sanz Serna & Calvo, 1994), we know that the phase space volume of (2) as well as of its symplectic approximation (6) is preserved. This gives rise to invertible model architectures leading to memory efficient implementations of the backpropagation through time algorithm, similar as in (Rusch & Mishra, 2021b). This denotes the most significant difference between the two different discretization schemes, i.e., the IMEX integration-based model is volume preserving, while IM integration-based model introduces dissipative terms. We will see in subsequent sections that both models have their own individual advantages depending on the underlying data. Finally, we note that stable higher-order time integration schemes (such as higher-order symplectic splitting schemes) can be used in this context as well. An example of leveraging the second-order symplectic velocity Verlet method can be found in Appendix Section D.

### 2.4 FAST RECURRENCE VIA ASSOCIATIVE PARALLEL SCANS

Parallel (or associative) scans, first introduced in Kogge & Stone (1973) and reviewed in Blelloch (1990), offer a powerful method for drastically reducing the computational time of recurrent operations. These scans have previously been employed to enhance the training and inference speed of RNNs (Martin & Cundy, 2017; Kaul, 2020). This technique was later adapted for state-space models in Smith et al. (2023), becoming a crucial component in the development of many state-of-the-art sequence models, including Linear Recurrent Units (LRUs) (Orvieto et al., 2023) and Mamba models (Gu & Dao, 2023).

A parallel scan operates on a sequence $[x_1, \ldots, x_N]$ with a binary associative operation $\bullet$, i.e., an operation satisfying $(x \bullet y) \bullet z = x \bullet (y \bullet z)$ for instances $x, y$, and $z$, to return the sequence $[x_1, x_1 \bullet x_2, \ldots, x_1 \bullet x_2 \bullet \cdots \bullet x_N]$. Under certain assumptions, this operation can be performed in computational time proportional to $\lceil \log_2(N) \rceil$. This is in stark contrast to the computational time of serial recurrence that is proportional to $N$. It is straightforward to check that the following operation is associative:

$$(\mathbf{a}_1, \mathbf{a}_2) \bullet (\mathbf{b}_1, \mathbf{b}_2) = (\mathbf{b}_1 \odot \mathbf{a}_1, \mathbf{b}_1 \odot \mathbf{a}_2 + \mathbf{b}_2),$$

where $\odot, \bullet$ are the matrix-matrix and matrix-vector products for matrices $\mathbf{a}_1, \mathbf{b}_1$ and vectors $\mathbf{a}_2, \mathbf{b}_2$. Note that both products can be computed in $\mathcal{O}(m)$ time leveraging $2 \times 2$ block matrices with only diagonal entries in each block, e.g., matrices $\mathbf{M}^{\text{IM}}$ and $\mathbf{M}^{\text{IMEX}}$ in (4),(6). Clearly, applying a parallel scan based on this associative operation on the input sequence $[(\mathbf{M}, \mathbf{F}_1), (\mathbf{M}, \mathbf{F}_2), \ldots]$ yields an output sequence $[(\mathbf{M}, \mathbf{F}_1), (\mathbf{M}^2, \mathbf{M}\mathbf{F}_1 + \mathbf{F}_2), \ldots]$ where the solution of the recurrent system $\mathbf{x}_n = \mathbf{M}\mathbf{x}_{n-1} + \mathbf{F}_n$ (with initial value $\mathbf{x}_0 = \mathbf{0}$) is stored in the second argument of the elements of this sequence. Thus, we can successfully apply parallel scans to both discretizations (4)(6) of our proposed system in order to significantly speed up computations. We refer to the application of this parallel scan to implicit formulations as implicit parallel scans. When applied to implicit-explicit formulations, we describe it as implicit-explicit parallel scans.

We term our proposed sequence model, which efficiently solves the underlying linear system of harmonic oscillators (2) using fast parallel scans, as **Lin**ear **O**scillatory **S**tate-**S**pace (**LinOSS**) model. To differentiate between the two discretization methods, we refer to the model derived from implicit discretization as **LinOSS-IM**, and the model based on the implicit-explicit discretization as **LinOSS-IMEX**. A schematic drawing of our proposed state-space model can be seen in Fig. 1. Moreover, a full description of a multi-layer LinOSS model, including specific nonlinear building blocks, can be found in Appendix A.

## 3 THEORETICAL INSIGHTS

### 3.1 STABILITY AND LEARNING LONG-RANGE INTERACTIONS

Hidden states of classical nonlinear RNNs are updated based on its previous hidden states pushed through a parametric function followed by a (usually) bounded nonlinear activation function such as tanh or sigmoid. This way, the hidden states are guaranteed to not blow up. However, this is not true for linear recurrences, where specific structures of the hidden-to-hidden weight matrix can lead to unstable dynamics. The following simple argument demonstrates that computing the eigenspectrum (i.e., set of eigenvalues) of the hidden-to-hidden weight matrix of a linear recurrent system suffices in order to analyse its stability properties. To this end, let us consider a general linear discrete dynamical system with external forcing given via the recurrence $\mathbf{x}_n = \mathbf{M}\mathbf{x}_{n-1} + \mathbf{B}\mathbf{u}_n$, and initial value $\mathbf{x}_0 = \mathbf{0}$. Assuming $\mathbf{M}$ is diagonalizable (if not, one can make a similar argument leveraging the Jordan normal form), i.e., there exists matrix $\mathbf{S}$ such that $\mathbf{M} = \mathbf{S}\mathbf{\Lambda}\mathbf{S}^{-1}$, where $\mathbf{\Lambda}$ is a diagonal matrix with the eigenvalues of $\mathbf{M}$ on its diagonal. The dynamics of the transformed hidden state $\bar{\mathbf{x}}_n = \mathbf{S}^{-1}\mathbf{x}_n$ evolve according to $\bar{\mathbf{x}}_n = \mathbf{S}^{-1}\mathbf{x}_n = \mathbf{S}^{-1}\mathbf{M}\mathbf{x}_{n-1} + \mathbf{S}^{-1}\mathbf{B}\mathbf{u}_n = \mathbf{\Lambda}\bar{\mathbf{x}}_{n-1} + \bar{\mathbf{B}}\mathbf{u}_n$ with initial value $\bar{\mathbf{x}}_0 = \mathbf{S}^{-1}\mathbf{x}_0$, where $\bar{\mathbf{B}} = \mathbf{S}^{-1}\mathbf{B}$. Unrolling the dynamics (assuming $\mathbf{x}_0 = \mathbf{0}$) yields,

$$\bar{\mathbf{x}}_1 = \bar{\mathbf{B}}\mathbf{u}_1, \quad \bar{\mathbf{x}}_2 = \mathbf{\Lambda}\bar{\mathbf{B}}\mathbf{u}_1 + \bar{\mathbf{B}}\mathbf{u}_2, \quad \ldots \quad \Rightarrow \mathbf{x}_n = \sum_{k=0}^{n-1} \mathbf{\Lambda}^k \bar{\mathbf{B}}\mathbf{u}_{n-k}.$$

Clearly, if all eigenvalues $\mathbf{\Lambda}$ have magnitude less or equal than 1 the hidden states will not blow up. In addition to stability guarantees, eigenspectra with unit norm allow the model to learn long-range

interactions (Arjovsky et al., 2016; Gu et al., 2022b; Orvieto et al., 2023) by avoiding vanishing and exploding gradients (Pascanu, 2013). Therefore, it is sufficient to analyse the eigenspectra of our proposed LinOSS models in order to understand their ability to generate stable dynamics and learn long-range interactions. To this end, we have the following propositions.

**Proposition 3.1.** *Let $\mathbf{M}^{IM} \in \mathbb{R}^{m \times m}$ be the hidden-to-hidden weight matrix of the implicit model LinOSS-IM (4). We assume that $\mathbf{A}_{kk} \geq 0$ for all diagonal elements $k = 1, \ldots, m$ of $\mathbf{A}$, and further that $\Delta t > 0$. Then, the complex eigenvalues of $\mathbf{M}^{IM}$ are given as,*

$$\lambda_j = \frac{1}{1 + \Delta t^2 \mathbf{A}_{kk}} + i(-1)^{\lceil \frac{j}{m} \rceil} \Delta t \frac{\sqrt{\mathbf{A}_{kk}}}{1 + \Delta t^2 \mathbf{A}_{kk}}, \quad \text{for all } j = 1, \ldots, 2m,$$

*with $k = j \bmod m$. Moreover, the spectral radius $\rho(\mathbf{M}^{IM})$ is bounded by $1$, i.e., $|\lambda_j| \leq 1$ for all $j = 1, \ldots, 2m$.*

*Proof.*

$$\det(\mathbf{M}^{IM} - \lambda \mathbf{I}) = \begin{vmatrix} \mathbf{S} - \lambda \mathbf{I} & -\Delta t \mathbf{A} \mathbf{S} \\ \Delta t \mathbf{S} & \mathbf{S} - \lambda \mathbf{I} \end{vmatrix} = \begin{vmatrix} \mathbf{S} - \lambda \mathbf{I} & -\Delta t \mathbf{A} \mathbf{S} \\ \mathbf{0} & \mathbf{S} - \lambda \mathbf{I} + \Delta t^2 \mathbf{A} \mathbf{S}^2 (\mathbf{S} - \lambda \mathbf{I})^{-1} \end{vmatrix}$$

$$= \prod_{k=1}^{m} (\mathbf{S}_{kk} - \lambda) \left( \mathbf{S}_{kk} - \lambda + \frac{\Delta t^2 \mathbf{A}_{kk} \mathbf{S}_{kk}^2}{\mathbf{S}_{kk} - \lambda} \right) = \prod_{k=1}^{m} [(\mathbf{S}_{kk} - \lambda)^2 + \Delta t^2 \mathbf{A}_{kk} \mathbf{S}_{kk}^2].$$

Setting $k = j \bmod m$, the eigenvalues of $\mathbf{M}^{IM}$ are thus given as,

$$\lambda_j = \mathbf{S}_{kk} + i(-1)^{\lceil \frac{j}{m} \rceil} \Delta t \mathbf{S}_{kk} \sqrt{\mathbf{A}_{kk}}, \quad \text{for all } j = 1, \ldots, 2m.$$

In particular, assuming $\mathbf{A}_{kk} \geq 0$ for all $k = 1, \ldots, m$, the magnitude of the eigenvalues $\lambda_j$ are given as,

$$|\lambda_j|^2 = \mathbf{S}_{kk}^2 + \Delta t^2 \mathbf{S}_{kk}^2 \mathbf{A}_{kk} = \mathbf{S}_{kk}^2 (1 + \Delta t^2 \mathbf{A}_{kk}) = \mathbf{S}_{kk} \leq 1,$$

for all $j = 1, \ldots, 2m$, with $k = j \bmod m$. $\qquad\square$

This proof reveals important insights into our proposed LinOSS-IM model. First, the magnitude of the eigenvalues at initialization can be controlled by either specific initialization of $\mathbf{A}$, or alternatively through a specific choice of the timestep parameter $\Delta t$. Moreover, LinOSS-IM yields asymptotically stable dynamics for any choices of positive parameters $\mathbf{A}$. This denotes a significant difference compared to previous first-order system, where the values of $\mathbf{A}$ (and thus its eigenvalues) have to be heavily constrained for the system to be stable. We argue that this flexibility in the parameterization of our model benefits the optimizer potentially leading to better performance in practice.

**Remark 1.** A straightforward adaptation of Proposition 3.1 can also be derived for the implicit-explicit version of the model, LinOSS-IMEX (6). The detailed proposition is given in Appendix Proposition E.1. In particular, the analysis shows that all eigenvalues $\lambda_j$ of $\mathbf{M}^{IMEX}$ in (6) satisfy $|\lambda_j| = 1$. This result underscores the key distinction between the two models: LinOSS-IM incorporates dissipative terms, whereas LinOSS-IMEX denotes a conservative system. Following the argument at the beginning of Section 3.1, one can interpret the dissipative terms in LinOSS-IM as forgetting mechanisms, which are considered crucial for expressive modeling of long sequences. This makes LinOSS-IM a more flexible model version compared to LinOSS-IMEX. Finally, we note that explicit discretization schemes lead to exploding hidden states returning NaN output values in practice during training. For this reason, we focus solely on stable methods in this context.

**Initialization and parameterization of weights.** Proposition 3.1 and Remark 1 reveal that both LinOSS models exhibit stable dynamics as long as the diagonal weights $\mathbf{A}$ in (1) are non-negative. This condition can easily be fulfilled via many different parameterizations of the diagonal matrix $\mathbf{A}$. An obvious choice is to square the diagonal values, i.e., a parameterization $\mathbf{A} = \hat{\mathbf{A}}\hat{\mathbf{A}}$, with diagonal matrix $\hat{\mathbf{A}} \in \mathbb{R}^{m \times m}$. Another straightforward approach is to apply the element-wise ReLU nonlinear activation function to $\mathbf{A}$, i.e., $\mathbf{A} = \text{ReLU}(\hat{\mathbf{A}})$, with $\text{ReLU}(x) = \max(0, x)$ and $\hat{\mathbf{A}}$ as before. The latter results in a LinOSS model where specific dimensions can be switched off completely by setting the corresponding weight in $\hat{\mathbf{A}}$ to a negative value. Due to this flexibility, we decide to focus on the ReLU-parameterization of the diagonal weights $\mathbf{A}$ in this manuscript.

After discussing the parameterization of $\mathbf{A}$, the next step is to determine an appropriate method for initializing its weights prior to training. Since we already know from Remark 1 that all absolute eigenvalues of the hidden matrix $\mathbf{M}^{\text{IMEX}}$ of LinOSS-IMEX are exactly one, the specific values of $\mathbf{A}$ are irrelevant for the model to learn long-range interactions. However, according to Proposition 3.1, $\mathbf{A}$ highly influences the eigenvalues of the hidden matrix $\mathbf{M}^{\text{IM}}$ of LinOSS-IM (4). As discussed in the previous section, high powers of the absolute eigenvalues of the hidden matrix are of particular interest when learning long-range interactions. To this end, we have the following Proposition concerning the expected powers of absolute eigenvalues for LinOSS-IM.

**Proposition 3.2.** *Let $\{\lambda_j\}_{j=1}^{2m}$ be the eigenspectrum of the hidden-to-hidden matrix $\mathbf{M}^{\text{IM}}$ of the LinOSS-IM model* (4). *We further initialize $\mathbf{A}_{kk} \sim \mathcal{U}([0, A_{max}])$ with $A_{max} > 0$ for all diagonal elements $k = 1, \ldots, m$ of $\mathbf{A}$ in* (2). *Then, the $N$-th moment of the magnitude of the eigenvalues are given as,*

$$\mathbb{E}(|\lambda_j|^N) = \frac{(\Delta t^2 A_{max} + 1)^{1 - \frac{N}{2}} - 1}{\Delta t^2 A_{max}(1 - \frac{N}{2})}, \tag{8}$$

*for all $j = 1, \ldots, 2m$, with $k = j \bmod m$.*

The proof, detailed in Appendix Section E.2, is a straight-forward application of the law of the unconscious statistician. Proposition 3.2 demonstrates that while the expectation of $|\lambda_i|^N$ might be much smaller than 1, it is still sufficiently large for practical use-cases. For instance, even considering $\Delta t = A_{\max} = 1$ and an extreme sequence length of $N = 100\text{k}$ still yields $\mathbb{E}(|\lambda_j|^N) = 1/49999 \approx 2 \times 10^{-5}$. Based on this and the fact that the values of $\mathbf{A}$ do not affect the eigenvalues for LinOSS-IMEX, we decide to initialize $\mathbf{A}$ according to $\mathbf{A}_{kk} \sim \mathcal{U}([0, 1])$ for both LinOSS models, while setting $\Delta t = 1$.

## 3.2 UNIVERSALITY OF LINOSS WITHIN CONTINUOUS AND CAUSAL OPERATORS

While trainability is an important aspect of learning long-range interactions, it does not demonstrate why LinOSS is able to express complex mappings between general (i.e., not necessarily oscillatory) input and output sequences. Therefore, in this section we analyze the approximation power of our proposed LinOSS model. Following the recent work of Lanthaler et al. (2024) we show that LinOSS is universal within the class of continuous and causal operators between time-series. To this end, we consider a full LinOSS block,

$$\mathbf{z}(t) = \mathbf{W}\sigma(\tilde{\mathbf{W}}\mathbf{y}(t) + \tilde{\mathbf{b}}), \tag{9}$$

with weights $\mathbf{W} \in \mathbb{R}^{q \times \tilde{m}}$, $\tilde{\mathbf{W}} \in \mathbb{R}^{\tilde{m} \times m}$, bias $\tilde{\mathbf{b}} \in \mathbb{R}^{\tilde{m}}$, element-wise nonlinear activation function $\sigma$ (e.g., $\tanh$ or ReLU), and solution $\mathbf{y}(t)$ of the LinOSS differential equations (2).

Based on this, we are now interested in approximating operators $\Phi : C_0([0, T]; \mathbb{R}^p) \to C_0([0, T]; \mathbb{R}^q)$ with the full LinOSS model (9), where

$$C_0([0, T]; \mathbb{R}^p) := \{\mathbf{u} : [0, T] \to \mathbb{R}^p \mid t \mapsto \mathbf{u}(t) \text{ is continuous and } \mathbf{u}(0) = \mathbf{0}\},$$

i.e., operators between continuous time-varying functions with values in $\mathbb{R}^p$. As pointed out in Lanthaler et al. (2024), the condition $\mathbf{u}(0) = \mathbf{0}$ is not restrictive and can directly be generalized to the case of any initial condition $\mathbf{u}(0) = \mathbf{u}_0 \in \mathbb{R}^p$ simply by introducing an arbitrarily small warm-up phase $[-t_0, 0]$ with $t_0 > 0$ of the oscillators to synchronize with the input signal $\mathbf{u}$. In addition to these function spaces, we pose the following conditions on the underlying operator $\Phi$,

1. $\Phi$ is *causal*, i.e., for any $t \in [0, T]$, if $\mathbf{u}, \mathbf{B} \in C_0([0, T]; \mathbb{R}^p)$ are two input signals, such that $\mathbf{u}|_{[0,t]} \equiv \mathbf{B}|_{[0,t]}$, then $\Phi(\mathbf{u})(t) = \Phi(\mathbf{B})(t)$.

2. $\Phi$ is *continuous* as an operator

$$\Phi : (C_0([0, T]; \mathbb{R}^p), \|\cdot\|_{L^\infty}) \to (C_0([0, T]; \mathbb{R}^q), \|\cdot\|_{L^\infty}),$$

with respect to the $L^\infty$-norm on the input-/output-signals.

**Theorem 3.3.** *Let $\Phi : C_0([0, T]; \mathbb{R}^p) \to C_0([0, T]; \mathbb{R}^q)$ be a causal and continuous operator. Let $K \subset C_0([0, T]; \mathbb{R}^p)$ be compact. Then for any $\epsilon > 0$, there exist hyperparameters $m$, $\tilde{m}$, diagonal weight matrix $\mathbf{A} \in \mathbb{R}^{m \times m}$, weights $\mathbf{B} \in \mathbb{R}^{m \times d}$, $\tilde{\mathbf{W}} \in \mathbb{R}^{\tilde{m} \times m}$, $\mathbf{W} \in \mathbb{R}^{q \times \tilde{m}}$ and bias vectors $\mathbf{b} \in \mathbb{R}^m$, $\tilde{\mathbf{b}} \in \mathbb{R}^{\tilde{m}}$, such that the output $\mathbf{z} : [0, T] \to \mathbb{R}^q$ of the LinOSS block (9) satisfies,*

$$\sup_{t \in [0, T]} |\Phi(\mathbf{u})(t) - \mathbf{z}(t)| \leq \epsilon, \quad \forall \mathbf{u} \in K.$$

The proof can be found in Appendix Section E.3. The main idea of the proof is to encode the infinite-dimensional operator $\Phi$ with a finite-dimensional operator that makes use of the structure of the LinOSS ODE system (1), and that can further be expressed by a (finite-dimensional) function. This theorem rigorously shows that LinOSS can approximate any causal and continuous operator between continuous time-varying functions with values in $\mathbb{R}^p$ to any desired accuracy.

## 4 EMPIRICAL RESULTS

In this section, we empirically test the performance of our proposed LinOSS models on a variety of challenging real-world sequential datasets, ranging from scientific datasets in genomics to practical applications in medicine. We ensure thereby a fair comparison to other state-of-the-art sequence models such as Mamba, LRU, and S5.

### 4.1 LEARNING LONG-RANGE INTERACTIONS

In the first part of the experiments, we focus on a recently proposed long-range sequential benchmark introduced in Walker et al. (2024). This benchmark focuses on six datasets from the University of East Anglia (UEA) Multivariate Time Series Classification Archive (UEA-MTSCA) (Bagnall et al., 2018), selecting those with the longest sequences for increased difficulty. The sequence lengths range thereby from $400$ to almost 18k. We compare our proposed LinOSS models to recent state-of-the-art sequence models, including state-space models such as Mamba and LRU. Table 1 shows the test accuracies averaged over five random model initialization and dataset splits of all six datasets for LinOSS as well as competing methods. We note that all other results are taken from Walker et al. (2024). Moreover, we highlight that we exactly follow the training procedure described in Walker et al. (2024) in order to ensure a fair comparison against competing models. More concretely, we use the same pre-selected random seeds for splitting the datasets into training, validation, and testing parts (using $70/15/15$ splits), as well as tune our model hyperparameters only on the same pre-described grid. In fact, since LinOSS does not possess any model specific hyperparameters – unlike competing state-space models such as LRU or S5 – our search grid is lower-dimensional compared to the other models considered. As a result, the hyperparameter tuning process involves significantly fewer model instances.

Table 1: Test accuracies averaged over $5$ training runs on UEA time-series classification datasets. All models are trained based on the same hyper-parameter tuning protocol in order to ensure fair comparability. The dataset names are abbreviations of the following UEA time-series datasets: Eigen-Worms (Worms), SelfRegulationSCP1 (SCP1), SelfRegulationSCP2 (SCP2), EthanolConcentration (Ethanol), Heartbeat, MotorImagery (Motor). The three best performing methods are highlighted in **red**[1] (First), **blue**[2] (Second), and **violet**[3] (Third).

| | Worms | SCP1 | SCP2 | Ethanol | Heartbeat | Motor | Avg |
|---|---|---|---|---|---|---|---|
| Seq. length | 17,984 | 896 | 1,152 | 1,751 | 405 | 3,000 | |
| #Classes | 5 | 2 | 2 | 4 | 2 | 2 | |
| NRDE | $83.9 \pm 7.3$ | $80.9 \pm 2.5$ | $\mathbf{53.7 \pm 6.9}^3$ | $25.3 \pm 1.8$ | $72.9 \pm 4.8$ | $47.0 \pm 5.7$ | 60.6 |
| NCDE | $75.0 \pm 3.9$ | $79.8 \pm 5.6$ | $53.0 \pm 2.8$ | $\mathbf{29.9 \pm 6.5}^2$ | $73.9 \pm 2.6$ | $49.5 \pm 2.8$ | 60.2 |
| Log-NCDE | $\mathbf{85.6 \pm 5.1}^3$ | $83.1 \pm 2.8$ | $\mathbf{53.7 \pm 4.1}^3$ | $\mathbf{34.4 \pm 6.4}^1$ | $75.2 \pm 4.6$ | $\mathbf{53.7 \pm 5.3}^3$ | $\mathbf{64.3}^3$ |
| LRU | $\mathbf{87.8 \pm 2.8}^2$ | $82.6 \pm 3.4$ | $51.2 \pm 3.6$ | $21.5 \pm 2.1$ | $\mathbf{78.4 \pm 6.7}^1$ | $48.4 \pm 5.0$ | 61.7 |
| S5 | $81.1 \pm 3.7$ | $\mathbf{89.9 \pm 4.6}^1$ | $50.5 \pm 2.6$ | $24.1 \pm 4.3$ | $\mathbf{77.7 \pm 5.5}^2$ | $47.7 \pm 5.5$ | 61.8 |
| S6 | $85.0 \pm 16.1$ | $82.8 \pm 2.7$ | $49.9 \pm 9.4$ | $26.4 \pm 6.4$ | $\mathbf{76.5 \pm 8.3}^3$ | $51.3 \pm 4.7$ | 62.0 |
| Mamba | $70.9 \pm 15.8$ | $80.7 \pm 1.4$ | $48.2 \pm 3.9$ | $\mathbf{27.9 \pm 4.5}^3$ | $76.2 \pm 3.8$ | $47.7 \pm 4.5$ | 58.6 |
| **LinOSS-IMEX** | $80.0 \pm 2.7$ | $\mathbf{87.5 \pm 4.0}^3$ | $\mathbf{58.9 \pm 8.1}^1$ | $\mathbf{29.9 \pm 1.0}^2$ | $75.5 \pm 4.3$ | $\mathbf{57.9 \pm 5.3}^2$ | $\mathbf{65.0}^2$ |
| **LinOSS-IM** | $\mathbf{95.0 \pm 4.4}^1$ | $\mathbf{87.8 \pm 2.6}^2$ | $\mathbf{58.2 \pm 6.9}^2$ | $\mathbf{29.9 \pm 0.6}^2$ | $75.8 \pm 3.7$ | $\mathbf{60.0 \pm 7.5}^1$ | $\mathbf{67.8}^1$ |

We can see in Table 1 that on average both LinOSS models outperform any other model we consider here by reaching an average (over all six datasets) accuracy of $65.0\%$ for LinOSS-IMEX and $67.8\%$ for LinOSS-IM. In particular, the average accuracy of LinOSS-IM is significantly higher than the two next best models, i.e., Log-NCDE reaching an average accuracy of $64.3\%$, and S6 reaching an accuracy of $62.0\%$ on average. It is particularly noteworthy that LinOSS-IM yields state-of-the-art results on the two datasets with the longest sequences, namely EigenWorms and MotorImagery.

## 4.2 VERY LONG-RANGE INTERACTIONS

In this experiment, we test the performance of LinOSS in the case of very long-range interactions. To this end, we consider the PPG-DaLiA dataset, a multivariate time series regression dataset designed for heart rate prediction using data collected from a wrist-worn device (Reiss et al., 2019). It includes recordings from fifteen individuals, each with approximately 150 minutes of data sampled at a maximum rate of 128 Hz. The dataset consists of six channels: blood volume pulse, electrodermal activity, body temperature, and three-axis acceleration. We follow Walker et al. (2024) and divide the data into training, validation, and test sets with a 70/15/15 split for each individual. After splitting the data, a sliding window of length 49920 and step size 4992 is applied. As in previous experiments, we apply the exact same hyperparameter tuning protocol to each model we consider here to ensure fair comparison. The test mean-squared error (MSE) of both LinOSS models as well as other competing models are shown in Table 2. We can see that both LinOSS models significantly outperform all other models. In particular, LinOSS-IM outperforms Mamba and LRU by nearly a factor of 2. This highlights the effectiveness of our proposed LinOSS models on sequential data with extreme length.

Table 2: Average test mean-squared error over 5 training runs on the PPG-DaLiA dataset. All models are trained following the same hyper-parameter tuning protocol in order to ensure fair comparability. The three best performing methods are highlighted in **red**[1] (First), **blue**[2] (Second), and **violet**[3] (Third).

| Model | MSE $\times 10^{-2}$ |
|---|---|
| NRDE (Morrill et al., 2021) | $9.90 \pm 0.97$ |
| NCDE (Kidger et al., 2020) | $13.54 \pm 0.69$ |
| Log-NCDE (Walker et al., 2024) | $\mathbf{9.56 \pm 0.59}$[3] |
| LRU (Orvieto et al., 2023) | $12.17 \pm 0.49$ |
| S5 (Smith et al., 2023) | $12.63 \pm 1.25$ |
| S6 (Gu & Dao, 2023) | $12.88 \pm 2.05$ |
| Mamba (Gu & Dao, 2023) | $10.65 \pm 2.20$ |
| **LinOSS-IMEX** | $\mathbf{7.5 \pm 0.46}$[2] |
| **LinOSS-IM** | $\mathbf{6.4 \pm 0.23}$[1] |

## 4.3 LONG-HORIZON FORECASTING

Inspired by Gu et al. (2021), we test our proposed LinOSS on its ability to serve as a general sequence-to-sequence model, even with weak inductive bias. To this end, we focus on time-series forecasting which typically requires specialized domain-specific preprocessing and architectures. We do, however, not alter our LinOSS model nor incorporate any inductive biases. Thus, we simply follow Gu et al. (2021) by setting up LinOSS as a general sequence-to-sequence model that treats forecasting as a masked sequence-to-sequence transformation. We consider a weather prediction task introduced in Zhou et al. (2021). In this task, several climate variables are predicted into the future based on local climatological data. Here, we focus on the hardest task in Zhou et al. (2021) of predicting the future 720 timesteps (hours) based on the past 720 timesteps. Table 3 shows the mean absolute error for both LinOSS models as well as other competing models. We can see that both LinOSS models outperform Transformers-based baselines as well as the other state-space models.

## 4.4 ADDITIONAL EXPERIMENTS AND ABLATIONS

Our proposed LinOSS model is the result of several design choices, such as the state matrix parameterization, state matrix initialization, and the numerical value of the discretization timestep $\Delta t$. In this section, we empirically analyze how these choices affect the performance of LinOSS by providing ablations and sensitivity studies.

We start by evaluating the performance of LinOSS under different parameterizations and initializations of the state matrix $\mathbf{A}$. Specifically, we examine LinOSS-IM within the experimental framework described in Section 4.1. For this analysis, we parameterize $\mathbf{A}$ using one of the two approaches proposed in Section 3.1: $\mathbf{A} = \tilde{\mathbf{A}}\tilde{\mathbf{A}}$ or $\mathbf{A} = \mathrm{ReLU}(\tilde{\mathbf{A}})$, where $\tilde{\mathbf{A}} \in \mathbb{R}^{m \times m}$ is a diagonal matrix. The

Table 3: Mean absolute error on the weather dataset predicting the future 720 time steps based on the past 720 time steps. The three best performing methods are highlighted in **red**[1] (First), **blue**[2] (Second), and **violet**[3] (Third).

| Model | Mean Absolute Error |
|---|---|
| Informer (Zhou et al., 2021) | 0.731 |
| LogTrans (Li et al., 2019) | 0.773 |
| Reformer (Kitaev et al., 2020) | 1.575 |
| LSTMa (Bahdanau et al., 2016) | 1.109 |
| LSTnet (Lai et al., 2018) | 0.757 |
| S4 (Gu et al., 2021) | 0.578[3] |
| **LinOSS-IMEX** | 0.508[1] |
| **LinOSS-IM** | 0.528[2] |

results, presented in Appendix Table 6 show that the ReLU parameterization leads to slightly better performance on average over all six datasets. However, the squared parameterization yields better performance on three out of six datasets. Thus, including different state matrix parameterization choices in the hyperparameter optimization can help achieve improved performance. We further test the performance of LinOSS-IM using a standard normal random initialization of the state matrix instead of the uniform initialization. The results of the standard normal initialization of the state matrix are shown in Appendix Table 6. We can see that this initialization yields similar performance to a uniform initialization on almost all considered datasets, except the EigenWorms dataset, where it obtains a lower mean test accuracy and a much higher standard deviation. This indicates that the performance of LinOSS-IM with a standard normal initialization for the state matrix is highly sensitive to the random seed used during initialization. This sensitivity arises because normal distributions do not have bounded support, allowing for the possibility of large matrix entries. These large entries result in small eigenvalues, which in turn lead to vanishing gradients.

Another natural question in this context concerns the sensitivity of performance to variations in the timestep $\Delta t$ used in the underlying discretization scheme. To investigate this, we train several LinOSS-IM models using different values of $\Delta t$ spanning three orders of magnitude, i.e., ranging from $10^{-3}$ to 1. The average test error, along with the standard deviation, is plotted for three different datasets in Appendix C.3. We can see that while the choice of $\Delta t$ does influence performance, the variations are not substantial.

Finally, we empirically analyze the roles of dissipation and conservation in LinOSS-IM and LinOSS-IMEX. As rigorously demonstrated in Section 3.1, LinOSS-IM introduces dissipative terms, whereas LinOSS-IMEX denotes a conservative system. While our earlier experiments examined the differences between these models using real-world data, we now focus on their performance for predicting energy-conserving dynamical systems to highlight their contrasting behaviors. To this end, we simulate the energy-conserving simple harmonic motion with various initial positions and velocities. The test error over time for both models is plotted in Appendix Fig. 2. The results show that the error in LinOSS-IM grows over time, whereas the error in LinOSS-IMEX remains constant. Notably, by the end of the time interval, LinOSS-IMEX outperforms LinOSS-IM by a factor of more than 8. This stark contrast underscores the fundamental difference between the two models: LinOSS-IM introduces dissipative terms, making it more effective for dissipative systems, while LinOSS-IMEX, being fully conservative, excels in energy-conserving systems.

## 5 CONCLUSION

In this paper, we introduce LinOSS, a state-space model based on harmonic oscillators. We rigorously show that LinOSS produces stable dynamics and is able to learn long-range interactions only requiring a nonnegative diagonal state matrix. In addition, we connect the underlying ODE system of the LinOSS model to Hamiltonian dynamics, which, discretized using symplectic integrators, perfectly preserves its symmetry of time reversibility. Moreover, we show that LinOSS is a universal approximator of continuous and causal operators between time-series. Together, these properties enable efficient modeling of long-range interactions, while ensuring stable and accurate long-horizon forecasting. Finally, we demonstrate that LinOSS outperforms state-of-the-art state-space models, such as Mamba, LRU, and S5.

ACKNOWLEDGMENTS

The authors would like to thank Dr. Alaa Maalouf (MIT) and Dr. T. Anderson Keller (Harvard University) for their insightful feedback and constructive suggestions on an earlier version of the manuscript. This work was supported in part by the Postdoc.Mobility grant P500PT-217915 from the Swiss National Science Foundation, the Schmidt AI2050 program (grant G-22-63172), and the Department of the Air Force Artificial Intelligence Accelerator and was accomplished under Cooperative Agreement Number FA8750-19-2-1000. The views and conclusions contained in this document are those of the authors and should not be interpreted as representing the official policies, either expressed or implied, of the Department of the Air Force or the U.S. Government. The U.S. Government is authorized to reproduce and distribute reprints for Government purposes notwithstanding any copyright notation herein.

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

**Supplementary Material for:**
Oscillatory State-Space Models

## A FULL ARCHITECTURE DETAILS

In this section, we outline the detailed architecture of a full multi-layer LinOSS model. To this end, the full LinOSS model starts by encoding an input sequence $\mathbf{u} = [\mathbf{u}_1, \mathbf{u}_2, \ldots, \mathbf{u}_N]$ with $\mathbf{u}_i \in \mathbb{R}^q$ for all $i = 1, \ldots, N$ via an affine transformation. After that, several blocks are applied consisting of a LinOSS layer (i.e., solving (1) with either IM or IMEX associative parallel scans) directly followed by a nonlinear layer using the Gaussian error linear unit activation function (GELU) (Hendrycks & Gimpel, 2023), the Gated Linear Unit (GLU) (Dauphin et al., 2017), i.e., $\mathrm{GLU}(\mathbf{x}) = \mathrm{sigmoid}(\mathbf{W}_1\mathbf{x}) \circ \mathbf{W}_2\mathbf{x}$, where $\mathbf{W}_{1,2}$ are learnable weight matrices, and a skip connection. Finally, the output of the final LinOSS block gets decoded by an affine transformation. The full LinOSS model is further presented in Algorithm 1. Note that weight matrices and bias vectors are applied parallel in time whenever possible (i.e., outside the recurrence). We are thus omitting subscript $i$ in Algorithm 1.

---

**Algorithm 1** Full LinOSS model

---

**Input:** Input sequence $\mathbf{u}$
**Output:** $L$-block LinOSS output sequence $\mathbf{o}$
    $\mathbf{u}^0 \leftarrow \mathbf{W}_{\mathrm{enc}}\mathbf{u} + \mathbf{b}_{\mathrm{enc}}$                                       ▷ Encode input sequence
    **for** $l = 1, \ldots, L$ **do**
        $\mathbf{y}^l \leftarrow$ solution of ODE in (1) with input $\mathbf{u}^{l-1}$ via parallel scan
        $\mathbf{x}^l \leftarrow \mathbf{C}\mathbf{y}^l + \mathbf{D}\mathbf{u}^{l-1}$                               ▷ Linear readout in (1)
        $\mathbf{x}^l \leftarrow \mathrm{GELU}(\mathbf{x}^l)$
        $\mathbf{u}^l \leftarrow \mathrm{GLU}(\mathbf{x}^l) + \mathbf{u}^{l-1}$
    **end for**
    $\mathbf{o} \leftarrow \mathbf{W}_{\mathrm{dec}}\mathbf{y}^L + \mathbf{b}_{\mathrm{dec}}$                        ▷ Decode final LinOSS block output

---

### A.1 SEQUENCE-TO-SEQUENCE LINOSS ARCHITECTURE FOR TIME-SERIES FORECASTING

While in sequence regression and sequence classification tasks we simply take the final output of the full LinOSS model at final time $T$ as the model prediction, we have to slightly adapt our architecture to handle time-series forecasting problems. To this end, we follow common practice for state-space models suggested in Gu et al. (2021) and generate train, validation, and test sequences of length $L_1 + L_2$, where $L_1$ is the number of the past sequence entries used for forecasting and $L_2$ are the number of future steps we aim to predict. Note that for the input sequences, we simply mask out the last $L_2$ entries of the sequence (i.e., with zero entries). Moreover, for the LinOSS output sequence, we only use the final $L_2$ entries for the future predictions.

## B TRAINING DETAILS

The code to run the experiments is implemented using the JAX auto-differentiation framework (Bradbury et al., 2018). All experiments were conducted on Nvidia Tesla V100 GPUs and Nvidia RTX 4090 GPUs, with the exception of the PPG experiment, which was run on Nvidia Tesla A100 GPUs due to higher memory demands.

### B.1 HYPERPARAMETERS

The hyperparameters of the models were optimized with the same grid search approach from Walker et al. (2024) for the six datasets in Section 4.1 and the PPG dataset of Section 4.2 to ensure perfect comparability with competing methods, i.e., using the grid: learning rate = $\{0.00001, 0.0001, 0.001\}$, number of layers = $\{2, 4, 6\}$, number of hidden neurons = $\{16, 64, 128\}$, state-space dimension = $\{16, 64, 256\}$, include time dimension $\{\mathrm{True}, \mathrm{False}\}$. Note that for the weather dataset, we performed a random search instead of grid search using the same hyperparameter bounds as before, except that we increased the maximum number of LinOSS blocks from 6 to 8.

The hyperparameters yielding the best performance can be seen in Table 4 for both LinOSS-IM and LinOSS-IMEX for each experiment in the main paper.

Table 4: Best hyperparameters

|  | Model | lr | hidden dim | state dim | #blocks | include time |
|---|---|---|---|---|---|---|
| **Worms** | IM | 0.001 | 128 | 64 | 2 | True |
|  | IMEX | 0.0001 | 64 | 16 | 2 | False |
| **SCP1** | IM | 0.0001 | 128 | 256 | 6 | True |
|  | IMEX | 0.0001 | 64 | 256 | 6 | False |
| **SCP2** | IM | 0.0001 | 128 | 64 | 6 | False |
|  | IMEX | 0.00001 | 64 | 256 | 6 | True |
| **Ethanol** | IM | 0.00001 | 16 | 16 | 4 | False |
|  | IMEX | 0.00001 | 16 | 256 | 4 | False |
| **Heartbeat** | IM | 0.001 | 16 | 16 | 6 | True |
|  | IMEX | 0.00001 | 64 | 16 | 2 | True |
| **Motor** | IM | 0.00001 | 128 | 16 | 2 | False |
|  | IMEX | 0.0001 | 16 | 256 | 6 | True |
| **PPG** | IM | 0.001 | 16 | 64 | 6 | True |
|  | IMEX | 0.0001 | 64 | 16 | 2 | True |
| **Weather** | IM | 0.0006 | 64 | 32 | 8 | True |
|  | IMEX | 0.0007 | 256 | 128 | 5 | False |

### B.2 MEMORY REQUIREMENTS, RUNTIMES, AND NUMBER OF PARAMETERS

In this section, we present the number of parameters, GPU memory usage (in MB), and run time (in seconds) for every considered model on all datasets from Section 4.1. The GPU memory and run time results for the other models are taken from Walker et al. (2024). Note that we used exactly the same GPU architecture as well as the same code and python libraries as in Walker et al. (2024) to ensure fair comparability, i.e., GPU memory usage and run time was measured on an Nvidia RTX 4090 GPU for all models. The run time was measured as the average run time for 1000 training steps. Table 5 comprehensively shows the number of parameters, GPU memory usage, and run time for all models. Note that we further follow Walker et al. (2024) and report these results for the best performing model identified during the previously described hyperparameter optimization process. Both LinOSS models exhibit comparable GPU memory usage and run time performance to other state-space models. Notably, LinOSS achieves the fastest runtime on two out of six datasets and ranks as the second fastest on another two datasets.

## C ADDITIONAL EXPERIMENTS

### C.1 ON THE STATE MATRIX PARAMETERIZATION AND INITIALIZATION

In the main paper, we have focused on parameterizing the state matrix $\mathbf{A}$ in (1) according to $\mathbf{A} = \mathrm{ReLU}(\hat{\mathbf{A}})$, with diagonal matrix $\hat{\mathbf{A}} \in \mathbb{R}^{m \times m}$. This was due to the fact that LinOSS requires the state matrix to be nonnegative in order to produce stable dynamics. However, another viable parameterization choice would be $\mathbf{A} = \hat{\mathbf{A}}\hat{\mathbf{A}}$. In this section, we test how the squared parameterization influences the performance of LinOSS on six long-range datasets taken from Section 4.1 of the main paper. Table 6 shows the average test accuracies of LinOSS-IM using a ReLU parameterization as well as using a squared parameterization together with the same baselines taken from the main paper. We can see that on average, the ReLU parameterization performs better. However, since the squared parameterization performs better on SCP2, Ethanol and Heartbeat (i.e., on three out of six datasets), we conclude that including the two parameterization choices in the hyperparameter optimization process will lead to even better performance.

Table 5: Number of parameters, GPU memory usage (in MB) and run time (in seconds) for every considered model on all long-range datasets from Section 4.1.

| | | NRDE | NCDE | Log-NCDE | LRU | S5 | Mamba | S6 | LinOSS-IMEX | LinOSS-IM |
|---|---|---|---|---|---|---|---|---|---|---|
| **Worms** | #parameters | 105110 | 166789 | 37977 | 101129 | 22007 | 27381 | 15045 | 26119 | 134279 |
| | GPU memory (MB) | 2506 | 2484 | 2510 | 10716 | 6646 | 13486 | 7922 | 6556 | 10654 |
| | run time (s) | 5386 | 24595 | 1956 | 94 | 31 | 122 | 68 | 37 | 90 |
| **SCP1** | #parameters | 117187 | 166274 | 91557 | 25892 | 226328 | 184194 | 24898 | 447944 | 991240 |
| | GPU memory (MB) | 716 | 694 | 724 | 960 | 1798 | 1110 | 904 | 4768 | 4772 |
| | run time (s) | 1014 | 973 | 635 | 9 | 17 | 7 | 3 | 42 | 38 |
| **SCP2** | #parameters | 200707 | 182914 | 36379 | 26020 | 5652 | 356290 | 26018 | 448072 | 399112 |
| | GPU memory (MB) | 712 | 692 | 714 | 954 | 762 | 2460 | 1222 | 4772 | 2724 |
| | run time (s) | 1404 | 1251 | 583 | 9 | 9 | 32 | 7 | 55 | 22 |
| **Ethanol** | #parameters | 93212 | 133252 | 31452 | 76522 | 76214 | 1032772 | 5780 | 70088 | 6728 |
| | GPU memory (MB) | 712 | 692 | 710 | 1988 | 1520 | 4876 | 938 | 4766 | 1182 |
| | run time (s) | 2256 | 2217 | 2056 | 16 | 9 | 255 | 4 | 48 | 8 |
| **Heartbeat** | #parameters | 15657742 | 1098114 | 168320 | 338820 | 158310 | 1034242 | 6674 | 29444 | 10936 |
| | GPU memory (MB) | 6860 | 1462 | 2774 | 1466 | 1548 | 1650 | 606 | 922 | 928 |
| | run time (s) | 9539 | 1177 | 826 | 8 | 11 | 34 | 4 | 4 | 7 |
| **Motor** | #parameters | 1134395 | 186962 | 81391 | 107544 | 17496 | 228226 | 52802 | 106024 | 91844 |
| | GPU memory (MB) | 4552 | 4534 | 4566 | 8646 | 4616 | 3120 | 4056 | 12708 | 4510 |
| | run time (s) | 7616 | 3778 | 730 | 51 | 16 | 35 | 34 | 128 | 11 |

Table 6: Test accuracies averaged over 5 training runs on UEA time-series classification datasets. All models are trained based on the same hyper-parameter tuning protocol in order to ensure fair comparability. The dataset names are abbreviations of the following UEA time-series datasets: EigenWorms (Worms), SelfRegulationSCP1 (SCP1), SelfRegulationSCP2 (SCP2), EthanolConcentration (Ethanol), Heartbeat, MotorImagery (Motor).

| | Worms | SCP1 | SCP2 | Ethanol | Heartbeat | Motor | Avg |
|---|---|---|---|---|---|---|---|
| Seq. length | 17,984 | 896 | 1,152 | 1,751 | 405 | 3,000 | |
| #Classes | 5 | 2 | 2 | 4 | 2 | 2 | |
| NRDE | $83.9 \pm 7.3$ | $80.9 \pm 2.5$ | $53.7 \pm 6.9$ | $25.3 \pm 1.8$ | $72.9 \pm 4.8$ | $47.0 \pm 5.7$ | 60.6 |
| NCDE | $75.0 \pm 3.9$ | $79.8 \pm 5.6$ | $53.0 \pm 2.8$ | $29.9 \pm 6.5$ | $73.9 \pm 2.6$ | $49.5 \pm 2.8$ | 60.2 |
| Log-NCDE | $85.6 \pm 5.1$ | $83.1 \pm 2.8$ | $53.7 \pm 4.1$ | $34.4 \pm 6.4$ | $75.2 \pm 4.6$ | $53.7 \pm 5.3$ | 64.3 |
| LRU | $87.8 \pm 2.8$ | $82.6 \pm 3.4$ | $51.2 \pm 3.6$ | $21.5 \pm 2.1$ | $78.4 \pm 6.7$ | $48.4 \pm 5.0$ | 61.7 |
| S5 | $81.1 \pm 3.7$ | $89.9 \pm 4.6$ | $50.5 \pm 2.6$ | $24.1 \pm 4.3$ | $77.7 \pm 5.5$ | $47.7 \pm 5.5$ | 61.8 |
| S6 | $85.0 \pm 16.1$ | $82.8 \pm 2.7$ | $49.9 \pm 9.4$ | $26.4 \pm 6.4$ | $76.5 \pm 8.3$ | $51.3 \pm 4.7$ | 62.0 |
| Mamba | $70.9 \pm 15.8$ | $80.7 \pm 1.4$ | $48.2 \pm 3.9$ | $27.9 \pm 4.5$ | $76.2 \pm 3.8$ | $47.7 \pm 4.5$ | 58.6 |
| **LinOSS-IM** (ReLU) | $95.0 \pm 4.4$ | $87.8 \pm 2.6$ | $58.2 \pm 6.9$ | $29.9 \pm 0.6$ | $75.8 \pm 3.7$ | $60.0 \pm 7.5$ | 67.8 |
| **LinOSS-IM** (squared) | $88.9 \pm 2.5$ | $86.6 \pm 1.8$ | $59.3 \pm 7.8$ | $32.7 \pm 6.2$ | $76.8 \pm 2.2$ | $58.2 \pm 8.4$ | 67.1 |
| **LinOSS-IM** (squared + Gaussian init) | $74.4 \pm 31.3$ | $87.3 \pm 1.7$ | $60.7 \pm 4.1$ | $31.4 \pm 4.8$ | $73.9 \pm 4.0$ | $56.5 \pm 3.0$ | 64.0 |

Another important question arises in the context of the state matrix initialization. While we argue in the main paper that initializing the state matrix using a simple random uniform distribution in $[0, 1]$ leads to models that are able to learn long-range interactions, we are interested in exploring other choices in this context. To this end, we train LinOSS-IM models and initialize their state matrix using a standard normal distribution. Note that we need to use the squared parameterization in this case, as ReLU would lead to switching off approximately half of the dimensions. The results for the six datasets from Section 4.1 of the main paper are shown in Table 6. We can see that initializing the state matrix using standard normal distribution leads to competitive results on almost all datasets except EigenWorms. Note that the standard deviation is very high on this dataset, suggesting that the performance is very sensitive to the random seed of the initialization. Therefore, the subpar performance on EigenWorms can be explained by the possibility of this initialization to produce large matrix entries that lead to small eigenvalues and thus vanishing gradients.

## C.2 DISSIPATIVE VS CONSERVATIVE LINOSS MODELS

In this section, we empirically analyze the dissipative behavior of LinOSS-IM and compare it to the conservative behavior of LinOSS-IMEX. To this end, we aim to predict an energy-conserving dynamical system. More concretely, we train our two LinOSS models to predict simple harmonic motion for different initial positions and velocities, i.e., the solution of,

$$
\begin{aligned}
y''(t) &= y(t), \\
y(0) &= A, \quad y'(0) = B,
\end{aligned}
\tag{10}
$$

for $A, B \in [0, 1]$. We construct train, validation, and test sets by solving (10) for uniform randomly chosen $A, B$, with 2000 train sequences, 500 validation sequences, and 500 test sequences. We set the stepsize for solving $y(t)$ to $\Delta t = 0.1$ and predict $y(t)$ for 1000 steps, i.e., for the time interval $[0, 100]$. Clearly, (10) is energy-conserving with the Hamiltonian given as $H(y, y') = y^2 + y'^2$.

Since state-space models are sequence-to-sequence models, we follow common practice and construct the input sequences as two-dimensional sequences of length 1000, where all entries of the first dimension are set to $A$ and the second dimension is set to $B$. The resulting test mean-squared error (MSE) is shown in Fig. 2 for each point in time. We can see that LinOSS-IMEX keeps the error constant, while the error for LinOSS-IM grows over time. This can be explained by the dissipative nature of LinOSS-IM, which forces the predicted trajectories to slowly converge to a steady-state, i.e., LinOSS-IM would go to zero in the asymptotic case of $t \to \infty$.

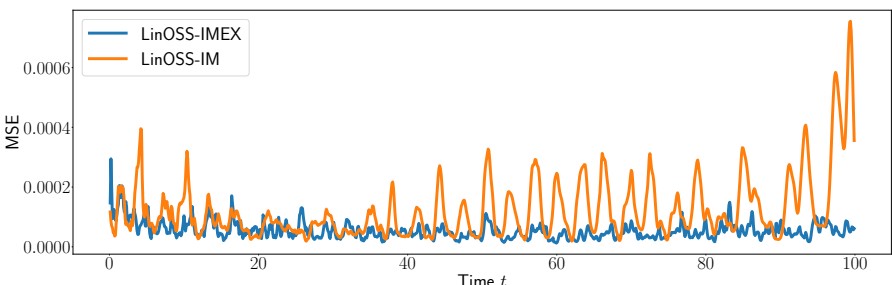

Figure 2: Mean squared error over time of LinOSS-IMEX and LinOSS-IM on predicting simple harmonic motion for different initial positions and velocities.

## C.3 ON THE SENSITIVITY OF $\Delta t$ IN LINOSS

While we set the timestep $\Delta t$ of the underlying time integration schemes to $\Delta t = 1$ for all our experiments in this paper, it is natural to ask whether different choices of $\Delta t$ will lead to different performance. To analyze this, we train LinOSS-IM models on three datasets from Section 4.1 of the main paper, i.e., SelfRegulationSCP1, Heartbeat, and MotorImagery dataset. We further vary $\Delta t$ between $10^{-3}$ and 1, i.e., spanning three orders of magnitude. We plot the average test accuracy (with standard deviation) in Fig. 3 for all three datasets. From this, we can conclude that while the choice of $\Delta t$ does influence performance, the variations are not substantial.

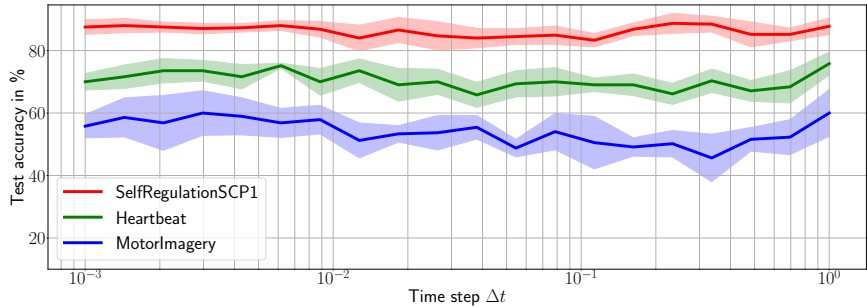

Figure 3: Test accuracies (mean and standard deviation over five different seeds) of LinOSS-IM on three different datasets from Section 4 of the main paper, i.e., SelfRegulationSCP1, Heartbeat, and MotorImagery, for varying values of the timestep $\Delta t$ of the underlying implicit time integration scheme of LinOSS (4).

## D    HIGHER-ORDER INTEGRATION SCHEMES

As outlined in the main text, higher-order discretization schemes can be readily applied in the context of oscillatory state-space models. Here, we provide an example on how to leverage the second-order velocity Verlet method as the underlying discretization method of LinOSS. To this end, applying the velocity Verlet integration scheme to our underlying system of ODEs (2) yields,

$$\mathbf{y}_n = \mathbf{y}_{n-1} + \Delta t \mathbf{z}_{n-1} + \frac{\Delta t^2}{2}(-\mathbf{A}\mathbf{y}_{n-1} + \mathbf{B}\mathbf{u}_n),$$

$$\mathbf{z}_n = \mathbf{z}_{n-1} + \frac{\Delta t}{2}(-\mathbf{A}\mathbf{y}_n + \mathbf{B}\mathbf{u}_{n+1} - \mathbf{A}\mathbf{y}_{n-1} + \mathbf{B}\mathbf{u}_n),$$

This can be rewritten in matrix form as,

$$\mathbf{x}_n = \mathbf{M}^{\mathrm{VV}}\mathbf{x}_{n-1} + \mathbf{F}_n^{\mathrm{VV}}, \tag{11}$$

with $\mathbf{x}_n = [\mathbf{y}_n, \mathbf{z}_n]^\top$, and,

$$\mathbf{M}^{\mathrm{VV}} = \begin{bmatrix} \mathbf{I} - \frac{\Delta t^2}{2}\mathbf{A} & \Delta t \mathbf{I} \\ -\Delta t \mathbf{A}(\mathbf{I} - \frac{\Delta t^2}{4}\mathbf{A}) & \mathbf{I} - \frac{\Delta t^2}{2}\mathbf{A} \end{bmatrix}, \quad \mathbf{F}_n^{\mathrm{VV}} = \begin{bmatrix} \frac{\Delta t^2}{2}\mathbf{B}\mathbf{u}_n \\ -\frac{\Delta t^3}{4}\mathbf{A}\mathbf{B}\mathbf{u}_n + \frac{\Delta t}{2}\mathbf{B}(\mathbf{u}_{n+1} + \mathbf{u}_n) \end{bmatrix}.$$

Equation (11) can be efficiently solved using fast associative parallel scans, as described in Section 2.4, leading to an alternative model architecture we refer to as LinOSS-VV. The key distinction from the symplectic LinOSS-IMEX lies in the discretization order: LinOSS-VV employs a second-order scheme, yielding more accurate approximations of the underlying ODE system compared to the first-order IMEX approach. However, this comes at the cost of greater computational complexity. We aim to explore the role of higher-order discretization schemes within the LinOSS framework in future research.

## E    SUPPLEMENT TO THE THEORETICAL INSIGHTS

### E.1    EIGENSPECTRUM OF LINOSS-IMEX

**Proposition E.1.** *Let $\mathbf{M}^{IMEX} \in \mathbb{R}^{m \times m}$ be the hidden-to-hidden weight matrix of the implicit-explicit model LinOSS-IMEX* (6). *We assume that $\mathbf{A}_{kk} > 0$ for all diagonal elements $k = 1, \ldots, m$ of $\mathbf{A}$, and further that $0 < \Delta t \leq \max\limits_{k=1,\ldots,m}(\frac{2}{\sqrt{\mathbf{A}_{kk}}})$. Then, the eigenvalues of $\mathbf{M}^{IMEX}$ are given as,*

$$\lambda_j = \frac{1}{2}(2 - \Delta t^2 \mathbf{A}_{kk}) + i(-1)^{\lceil \frac{j}{m} \rceil}\frac{1}{2}\sqrt{\Delta t^2 \mathbf{A}_{kk}(4 - \Delta t^2 \mathbf{A}_{kk})}, \quad \text{for all } j = 1, \ldots, 2m,$$

*with $k = j \bmod m$. Moreover, all absolute eigenvalues of $\mathbf{M}^{IMEX}$ are exactly 1, i.e., $|\lambda_j| = 1$ for all $j = 1, \ldots, 2m$.*

*Proof.* Following the same procedure outlined in the proof of Proposition 3.1, the eigenvalues of $\mathbf{M}^{IMEX}$ are given as,

$$\lambda_j = \frac{1}{2}(2 - \Delta t^2 \mathbf{A}_{kk}) + i(-1)^{\lceil \frac{j}{m} \rceil}\frac{1}{2}\sqrt{\Delta t^2 \mathbf{A}_{kk}(4 - \Delta t^2 \mathbf{A}_{kk})}, \quad \text{for all } j = 1, \ldots, 2m,$$

with $k = j \bmod m$. To calculate their absolute value, we must consider two distinct cases.

1. $\underline{\Delta t^2 \mathbf{A}_{kk} = 4}$: it follows directly that $|\lambda_j|^2 = (\frac{1}{2}(-4+2))^2 = 1$.

2. $\underline{\Delta t^2 \mathbf{A}_{kk} < 4}$: With this assumption, we can compute the absolute value of $\lambda_j$ as,

$$|\lambda_j|^2 = \left(\frac{2 - \Delta t^2 \mathbf{A}_{kk}}{2}\right)^2 + \frac{\Delta t^2}{4}\mathbf{A}_{kk}(4 - \Delta t^2 \mathbf{A}_{kk})$$

$$= \frac{4 - 4\Delta t^2 \mathbf{A}_{kk} + \Delta t^4 \mathbf{A}_{kk}^2}{4} + \frac{4\Delta t^2}{4}\mathbf{A}_{kk} - \frac{\Delta t^4 \mathbf{A}_{kk}^2}{4}$$

$$= 1.$$

$\square$

### E.2 PROOF OF PROPOSITION 3.2

**Proposition.** *Let $\{\lambda_j\}_{j=1}^{2m}$ be the eigenspectrum of the hidden-to-hidden matrix $\mathbf{M}^{IM}$ of the LinOSS-IM model (4). We further initialize $\mathbf{A}_{kk} \sim \mathcal{U}([0, A_{max}])$ with $A_{max} > 0$ for all diagonal elements $k = 1, \ldots, m$ of $\mathbf{A}$ in (2). Then, the $N$-th moment of the magnitude of the eigenvalues are given as,*

$$\mathbb{E}(|\lambda_j|^N) = \frac{(\Delta t^2 A_{max} + 1)^{1-\frac{N}{2}} - 1}{\Delta t^2 A_{max}(1 - \frac{N}{2})}, \tag{12}$$

*for all $j = 1, \ldots, 2m$, with $k = j \bmod m$.*

*Proof.* By the law of the unconscious statistician together with the identity $|\lambda_j| = \sqrt{\mathbf{S}_k}$ from the proof of Proposition 3.1 it follows that,

$$\mathbb{E}(|\lambda_j|^N) = \frac{1}{A_{\max}} \int_0^{A_{\max}} (1 + \Delta t^2 x)^{-\frac{N}{2}} dx = \frac{1}{\Delta t^2 A_{\max}} \int_1^{\Delta t^2 A_{\max}+1} u^{-\frac{N}{2}} du$$

$$= \frac{(\Delta t^2 A_{\max} + 1)^{1-\frac{N}{2}} - 1}{\Delta t^2 A_{\max}(1 - \frac{N}{2})},$$

where we substituted $u = \Delta t^2 x + 1$. $\qquad\qquad\square$

### E.3 PROOF OF THEOREM 3.3

**Theorem.** *Let $\Phi : C_0([0, T]; \mathbb{R}^p) \to C_0([0, T]; \mathbb{R}^q)$ be a causal and continuous operator. Let $K \subset C_0([0, T]; \mathbb{R}^p)$ be compact. Then for any $\epsilon > 0$, there exist hyperparameters $m$, $\tilde{m}$, diagonal weight matrix $\mathbf{A} \in \mathbb{R}^{m \times m}$, weights $\mathbf{B} \in \mathbb{R}^{m \times d}$, $\tilde{\mathbf{W}} \in \mathbb{R}^{\tilde{m} \times m}$, $\mathbf{W} \in \mathbb{R}^{q \times \tilde{m}}$ and bias vectors $\mathbf{b} \in \mathbb{R}^m$, $\tilde{\mathbf{b}} \in \mathbb{R}^{\tilde{m}}$, such that the output $\mathbf{z} : [0, T] \to \mathbb{R}^q$ of the LinOSS model (9) satisfies,*

$$\sup_{t \in [0,T]} |\Phi(\mathbf{u})(t) - \mathbf{z}(t)| \leq \epsilon, \quad \forall \mathbf{u} \in K.$$

*Proof.* We begin by considering the simple forced harmonic oscillator,

$$\mathbf{y}''(t) = -A^2 \mathbf{y}(t) + \mathbf{u}(t), \tag{13}$$

where $\mathbf{u}(t) \in \mathbb{R}^p$ is an external forcing and we assume $A \neq 0$. We further introduce the time-windowed sine transform for the input signal $\mathbf{u}(t)$,

$$\mathcal{L}_t \mathbf{u}(A) = \int_0^t \mathbf{u}(t - \tau) \sin(A\tau) d\tau.$$

Then, a straightforward calculation shows that the solution $\mathbf{y}(t)$ to (13) computes (up to a constant) a time-windowed sine transform, i.e.,

$$y(t) = A^{-1} \int_0^t \mathbf{u}(\tau) \sin(A(t - \tau)) d\tau. \tag{14}$$

This can easily be verified by differentiating $\mathbf{y}$,

$$\mathbf{y}'(t) = \int_0^t \mathbf{u}(\tau) \cos(A(t - \tau)) d\tau + A^{-1}[\mathbf{u}(\tau) \sin(A(t - \tau))]_{\tau=t}$$

$$= \int_0^t \mathbf{u}(\tau) \cos(A(t - \tau)) d\tau.$$

Differentiating one more time yields,

$$\mathbf{y}''(t) = -A \int_0^t \mathbf{u}(\tau) \sin(A(t - \tau)) d\tau + [\mathbf{u}(\tau) \cos(A(t - \tau))]_{\tau=t}$$

$$= -A \int_0^t \mathbf{u}(\tau) \sin(A(t - \tau)) d\tau + \mathbf{u}(t)$$

$$= -A^2 \mathbf{y}(t) + \mathbf{u}(t).$$

We will now make use of the fundamental Lemma in Lanthaler et al. (2024) that provides a finite-dimensional encoding of the operator $\Phi$ we wish to approximate, i.e., for any $\epsilon > 0$, there exists $N \in \mathbb{N}$, weights $A_1, \ldots, A_N$ and a continuous mapping $\Psi : \mathbb{R}^{p \times N} \times [0, T^2/4] \to \mathbb{R}^q$, such that

$$|\Phi(\mathbf{u})(t) - \Psi(\mathcal{L}_t\mathbf{u}(A_1), \ldots, \mathcal{L}_t\mathbf{u}(A_N); t^2/4)| \le \epsilon,$$

for all $\mathbf{u} \in K$ and $t \in [0, T]$.

Using the result in (14) we can then construct the input vector $[\mathcal{L}_t\mathbf{u}(A_1), \ldots, \mathcal{L}_t\mathbf{u}(A_N), t^2/4]^\top \in \mathbb{R}^{pN} \times [0, T^2/4]$ based on the ODE system (1) underlying our LinOSS model:

$$\begin{bmatrix} \mathcal{L}_t\mathbf{u}(A_1) \\ \mathcal{L}_t\mathbf{u}(A_2) \\ \vdots \\ \mathcal{L}_t\mathbf{u}(A_N) \\ t^2/4 \end{bmatrix} = \tilde{\mathbf{A}}\mathbf{y}(t), \tag{15}$$

where $\mathbf{y}(t) \in \mathbb{R}^{pN}$ solves the system,

$$\mathbf{y}''(t) = -\mathbf{A}^2\mathbf{y} + \mathbf{B}\mathbf{u}(t) + \mathbf{b}, \tag{16}$$

with

$$\mathbf{A} = \text{diag}([\underbrace{A_1, \ldots, A_1}_{p-\text{times}}, \underbrace{A_2, \ldots, A_2}_{p-\text{times}}, \ldots \ldots, \underbrace{A_N, \ldots, A_N}_{p-\text{times}}, 0]),$$

$$\mathbf{B} = [\underbrace{\mathbf{I}_p, \ldots, \mathbf{I}_p}_{N-\text{times}}, 0]^\top, \quad \mathbf{b} = [0, \ldots, 0, 1/2]^\top$$

where $\mathbf{I}_p \in \mathbb{R}^{p \times p}$ is the identity matrix and $\tilde{\mathbf{A}}$ equals $\mathbf{A}$, except that $\tilde{\mathbf{A}}_{pN,pN} = 1$. Thus, $\tilde{\mathbf{A}}\mathbf{y}(t)$ computes exactly the input to the finite-dimensional operator $\Psi$. By the universal approximation theorem for ordinary neural networks there exist weight matrices $\mathbf{W}, \hat{\mathbf{W}}$ and bias $\tilde{\mathbf{b}}$, such that,

$$|\Psi(\mathcal{L}_t\mathbf{u}(A_1), \ldots, \mathcal{L}_t\mathbf{u}(A_N); t^2/4) - \mathbf{W}\sigma(\hat{\mathbf{W}}[\mathcal{L}_t\mathbf{u}(A_1), \ldots, \mathcal{L}_t\mathbf{u}(A_N), t^2/4]^\top + \tilde{\mathbf{b}})| < \epsilon.$$

Thus, for every $\mathbf{u} \in K$ we have,

$$\begin{aligned} |\Phi(\mathbf{u}(t)) - \mathbf{z}(t)| \le &|\Phi(\mathbf{u}(t)) - \Psi(\mathcal{L}_t\mathbf{u}(A_1), \ldots, \mathcal{L}_t\mathbf{u}(A_N); t^2/4)| \\ &+ |\Psi(\mathcal{L}_t\mathbf{u}(A_1), \ldots, \mathcal{L}_t\mathbf{u}(A_N); t^2/4) - \mathbf{W}\sigma(\tilde{\mathbf{W}}\mathbf{y}(t) + \tilde{\mathbf{b}})| \\ &< 2\epsilon, \end{aligned}$$

with $\tilde{\mathbf{W}} = \hat{\mathbf{W}}\tilde{\mathbf{A}}$. Since $\epsilon > 0$ was arbitrary, we conclude that for any causal and continuous operator $\Phi : C_0([0, T]; \mathbb{R}^p) \to C_0([0, T]; \mathbb{R}^q)$, compact set $K \subset C_0([0, T]; \mathbb{R}^p)$ and $\epsilon > 0$, there exists a LinOSS model of form (9), which uniformly approximates $\Phi$ to accuracy $\epsilon$ for all $\mathbf{u} \in K$. This completes the proof.

$\square$

**Remark 2.** A natural question arises regarding the performance of LinOSS and other state-space models when learning chaotic dynamical systems. A key issue in the context of learning chaotic systems with recurrent models is the inevitability of exploding gradients during training, as rigorously demonstrated in Mikhaeil et al. (2022). While our universality result, Theorem 3.3, still holds assuming the assumptions stated are fulfilled, it can be seen that the Lipschitz constant of the readout MLP (i.e., approximation of $\Psi$ in the proof of Theorem 3.3) would blow up, thereby enabling exploding gradients. However, it is crucial to emphasize that this issue persists with all recurrent models and is not unique to LinOSS or other state-space models. Interestingly, it has already been empirically shown in Hu et al. (2024) that state-space models (i.e., models based on linear dynamics) vastly outperform chaotic RNNs such as LSTMs (Hochreiter, 1997) and GRUs (Chung et al., 2014) for learning chaotic dynamical systems.

