# OpenReview forum: "Oscillatory State-Space Models"
_ICLR.cc/2025/Conference — ICLR 2025 Oral_

### Official Review · Reviewer_w9Vn · 2024-11-03

**Soundness:** 3
**Presentation:** 3
**Contribution:** 2
**Rating:** 8
**Confidence:** 3

**Summary:**

In this work, the authors propose a state-space model (SSM) architecture, Linear-Oscillatory SSM, derived from the discretization of second-order linear ODEs that models a network of forced harmonic oscillators. They develop two discretization schemes for the approach, study the stability properties of the parametrization and show universal approximation of LinOSS for approximating general continuous, causal input-output mapping. Through empirical evaluations, they demonstrate that their approach outperforms other Linear SSMs on time series classification, prediction and long-horizon forecasting.

**Strengths:**

The work is well motivated and the writing is clear and concise. The recurrence matrix afforded by the proposed approach has desirable stability properties and is less constrained compared to the typical SSM parametrizations. The expressivity is further supported by the theoretical analysis on universality of the proposed parametrization. The experimental results are thorough and demonstrate the efficacy of the proposed approach relative to baselines.

**Weaknesses:**

Given that the parametrization has been introduced in Rusch & Mishra (2021) and the universality results for the non-linear counterpart to this work have been shown in Lanthaler et al., 2024, the novelty of this work is somewhat limited. Still, I think that this is a useful contribution overall as it improves the ability of SSMs for learning long-range dependencies.

**Questions:**

* L116-118. Since the oscillators are independent, can the proposed architecture model transient synchronization/desynchronization?
* L250. `Assuming M is diagonalizable [...]`. If $M$ has real eigenvalues with algebraic multiplicity $ > 1$, would that make the system unstable? Admittedly, norm growth would be sub-exponential, so perhaps in practice it's fine.
* L289 `constraint` $=>$ constrained

---

> ### Author Response · Authors · 2024-11-21
> **Reply to Reviewer w9Vn**
>
> We thank the reviewer for appreciating the merits of our paper and for the insightful feedback. Below, we address the concerns raised by the reviewer.
>
> - **''novelty of this method''**
> We argue that the novelty of our work lies in the state-space formulation of oscillatory systems, introducing a fresh perspective to this domain. Specifically, we employ a system of linear oscillators, in contrast to the nonlinear systems explored by Rusch and Mishra (2021). Additionally, achieving an efficient state-space modeling approach required multiple innovations, enabling us to leverage fast associative parallel scans for these systems effectively. Moreover, the linear formulation of these systems allowed for explicit theoretical statements about these models, which is not available for previous nonlinear systems.
>
> - **''Since the oscillators are independent, can the proposed architecture model transient synchronization / desynchronization?''**
> This is a very interesting question. While the oscillators in each LinOSS layer are not connected, a notion of coupling can be introduced via stacking several layers, where the output of the previous layer is the input of the current layer and the matrix $\bf B$ modulating these inputs is dense, i.e., allows for mixing. Thus effects such as synchronization/desynchronization are possible. Another way to approach this would be to introduce sparse coupling within each layer and simply diagonalizing the resulting state matrix to use fast associative parallel scans. We plan to investigate this in more detail together with other important notions from computational neuroscience in future research.
>
> - **''If $\bf M$ has real eigenvalues with algebraic multiplicity
> $>1$, would that make the system unstable?''**
> An algebraic multiplicity bigger than 1 does not necessarily make the system unstable. One example for this is our LinOSS-IMEX model, which has algebraic multiplicity equal to the state dimension $m$. However, we show in Proposition E.1 that this system is stable.
>
> We sincerely hope that we have addressed the concerns of the reviewer satisfactorily in the revised version and would kindly ask the reviewer to update their score accordingly.

---

> > ### Comment · Reviewer_w9Vn · 2024-11-27
> >
> > I thank the reviewers for their response and rebuttal, and I agree that the application of linear oscillators to the SSM domain is novel. The revisions have made the overall paper stronger and I will be raising my score.

---

> > > ### Author Response · Authors · 2024-11-27
> > > **Thanking the Reviewer**
> > >
> > > We thank the reviewer for their positive assessment of our revised version and for raising our score.

---

### Official Review · Reviewer_Wn4t · 2024-11-04

**Soundness:** 3
**Presentation:** 3
**Contribution:** 3
**Rating:** 8
**Confidence:** 4

**Summary:**

This paper introduces Linear Oscillatory State-Space models (LinOSS), a novel approach to sequence modeling based on forced harmonic oscillators. The model comes in two variants: LinOSS-IM (implicit) and LinOSS-IMEX (implicit-explicit). The key innovation is using second-order ODEs with diagonal state matrices, offering stable dynamics while only requiring non-negative diagonal elements. The paper provides theoretical guarantees for stability and universal approximation, along with empirical validation showing significant improvements over state-of-the-art models like Mamba and LRU on long sequences.

**Strengths:**

The paper demonstrates strong theoretical foundations by providing rigorous mathematical analysis of stability conditions, proving universal approximation capability, and establishing clear connections to Hamiltonian systems and symplectic integration. Implementation-wise, it offers a remarkably simple parameterization requiring only non-negative diagonal elements, achieves efficient computation through parallel scans, and presents two complementary variants with different preservation properties. The empirical results are great, showing strong performance on diverse tasks.

**Weaknesses:**

1. Limited Analysis of Model Interpretability:

While based on oscillatory dynamics, lacks discussion of learned frequencies

No analysis of how the model captures different timescales

2. Experiment:

No ablation studies on the impact of different initialization schemes

The implementation details are unclear. For instance, how does it compare to Mamba or S5 in terms of speed, training time, FLOPs, and memory usage? Discussing these aspects could enhance its practical utility.

**Questions:**

1. Could the model be extended to incorporate coupled oscillations while maintaining stability guarantees?

2. What is the impact of the time step parameter Δt on model performance and stability?

3. How does the choice between IM and IMEX variants affect training dynamics and convergence?

---

> ### Author Response · Authors · 2024-11-21
> **Reply to Reviewer Wn4t**
>
> We thank the reviewer for appreciating the merits of our paper and their suggestions for improvement. Below, we address the concerns raised by the reviewer.
>
> - **''Limited Analysis of Model Interpretability''** We sincerely thank the reviewer for their insightful comment. We fully agree that exploring model interpretability is both important and intriguing. In response, we have added a new experiment in the revised version of our paper (see Section C.2 in the appendix) to analyze the physics underlying our model. Specifically, we predict simple harmonic motion for varying initial positions and velocities over an extended period. The results demonstrate that LinOSS-IMEX maintains a constant error, whereas the error for LinOSS-IM increases over time. This is due to the fact that LinOSS-IM introduces dissipative behavior and in the asymptotic limit would converge towards a steady-sate for any state matrix values. This is in contrast to the conservative nature of LinOSS-IMEX. We thank the reviewer once again for their valuable suggestion, which has significantly enhanced the interpretability aspect of our work.
>
> - **''No analysis of how the model captures different timescales''**
> We thank the reviewer for the comment. We would like to point out that most, if not all, our datasets are based on 'real-world' data and thus likely contain information arranged according to multiple scales. The strong performance of LinOSS on these dataset indicates that LinOSS is able to capture multiple scales in the underlying data. One highlight is the weather prediction task, which is known to be multiscale. Again, LinOSS outperforms any other competing method considered in this task, suggesting strong performance when applied to multiscale data. From a theoretical perspective, our universality result also holds for multiscale dynamical systems. Thus, we can show that LinOSS can approximate any multiscale system up to any desired accuracy.
>
> - **''No ablation studies on the impact of different initialization schemes''**
> We appreciate the reviewer's suggestion and agree with their observation. We have now added a new experiment (Section C.1), testing several different initializations of the state matrix. Additionally, we have provided a detailed discussion of the results in Section 4.4. The findings support our claim that uniform random initialization is a robust choice in this context. Thank you again for highlighting this important aspect.
>
> - **''The implementation details are unclear.''**
> We fully agree with the reviewer and  have now added run times, GPU memory usage, and number of parameters for all models on all six datasets from Section 4.1 in Table 5 in the Appendix. While LinOSS seems to be well within the efficiency range of other state-space models, we would like to highlight that LinOSS is the fastest among all other considered models on two out of six datasets, as well as the second fastest on again two out of six datasets.
>
> - **''Could the model be extended to incorporate coupled oscillations while maintaining stability guarantees?''**
> This is a very interesting question. Indeed, coupling within one LinOSS layer can be incorporated leveraging the diagonalization trick outlined in the first paragraph of Section 3.1. However, one would need to rely on a fast diagonalization of the underlying state matrix. That being said, the oscillators are already coupled in LinOSS, however, delayed by different layers. Since the output of the previous layer denotes the input of the current layer and due to the fact that the matrix $\bf B$ in equation (1) is dense, this introduces a notion of coupling.
>
> - **''What is the impact of the time step parameter $\Delta t$ on model performance and stability?''**
> We have added a proposition on the stability of LinOSS-IMEX in Section E.1. There, we have rigorously worked out the impact of $\Delta t$ on the stability of LinOSS. Moreover, motivated by the reviewer's suggestion, we have added a new experiment where we analyze the impact of $\Delta t$ on the performance of LinOSS (Section C.3 and Section 4.4). From our obtained results, we can conclude that while the choice of $\Delta t$ does influence performance, the variations are not substantial.
>
> - **''How does the choice between IM and IMEX variants affect training dynamics and convergence?''**
> This is a very interesting question. Based on our new experiments analyzing the underlying physics of our proposed LinOSS models, we argue that LinOSS-IM converges faster and has better generalization compared to LinOSS-IMEX on data that includes conserved quantities. However, whenever data exhibits dissipative behavior, or forgetting is crucial, the roles are swapped.
>
> We sincerely hope that we have addressed the concerns of the reviewer satisfactorily in the revised version and would kindly ask the reviewer to update their score accordingly.

---

> > ### Comment · Reviewer_Wn4t · 2024-11-27
> >
> > Thank you for addressing my concerns. I’ve updated my rating accordingly.

---

> > > ### Author Response · Authors · 2024-11-27
> > > **Thanking the Reviewer**
> > >
> > > We thank the reviewer for appreciating the merits of our revised version and rebuttal and for increasing our score.

---

### Official Review · Reviewer_su1u · 2024-11-04

**Soundness:** 4
**Presentation:** 3
**Contribution:** 4
**Rating:** 8
**Confidence:** 2

**Summary:**

The paper provides a novel state-space model. They have two different versions of it, in which they rigorously show the power of their algorithms and also experimentally verify it. The method outperforms SOTA methods in many tasks.

**Strengths:**

Provides strong theoretical together with intuitive explanations.
Contrasts their two proposed methods mathematically and also experimentally.
The experimental results are excellent and definitely contributes to the field significantly.
Supplementary material is comprehensive.

**Weaknesses:**

I think section 3.2 can be written more accessible.

I believe Figure 1 is very important but can be made more explanatory.

**Questions:**

To the best of my understanding, the model cannot produce chaotic dynamics. What if the task in hand requires this? How does this contrast (if there is a contrast) with section 3.2?

---

> ### Author Response · Authors · 2024-11-21
> **Reply to Reviewer su1u**
>
> We thank the reviewer for appreciating the merits of our paper and their suggestions for improvement. Below, we address the concerns raised by the reviewer.
>
> - **''I think section 3.2 can be written more accessible.''** We appreciate the reviewer’s feedback and have revised parts of Section 3.2 accordingly. Specifically, we have clarified abstract concepts, including the specific function spaces, by providing definitions in more straightforward terms. Moreover, we have summarized the main statement of Theorem 3.3.
>
> - **''I believe Figure 1 is very important but can be made more explanatory.''** We have added further explanations, e.g., labeling $\bf y(t)$ and $\bf u(t)$ in the figure. Moreover, we have added a new section in the appendix (Section A) that explains the full architecture in detail, including the nonlinear layers. We hope that this together with the changes in the figure made it more explanatory.
>
> - **''To the best of my understanding, the model cannot produce chaotic dynamics. What if the task in hand requires this?''** This is a very interesting question. Indeed, due to the inherent linear nature of state-space models one might expect that learning chaotic time-series using these models might denote a challenge. However, first experiments in this context suggest that state-space models actually perform well on chaotic dynamical systems (Hu et al., 2024), significantly outperforming their chaotic RNN counterparts such as LSTMs and GRUs. We further note that our universality result still holds. However, the Lipschitz constant in this case would need to blow up. That being said, this should not matter locally.
>
> Paper: Zheyuan Hu, Nazanin Ahmadi Daryakenari, Qianli Shen, Kenji Kawaguchi, George Em Karniadakis, "State-space models are accurate and efficient neural operators for dynamical systems". 2024.

---

> > ### Comment · Reviewer_su1u · 2024-11-22
> >
> > Thank you to the authors for their response.
> >
> > I’m a bit confused about the universality result. Hess et al. (2023) and Mikhaeil et al. (2022) show that “…the gradients of RNNs with chaotic dynamics always diverge.” I’m not an expert in this area but would appreciate some discussion on this point. (not necessarily to be added to the main script.)
> >
> > Hess, F., Monfared, Z., Brenner, M., & Durstewitz, D. (2023). Generalized teacher forcing for learning chaotic dynamics. arXiv preprint arXiv:2306.04406.
> >
> > Mikhaeil, J., Monfared, Z., & Durstewitz, D. (2022). On the difficulty of learning chaotic dynamics with RNNs. Advances in Neural Information Processing Systems, 35, 11297-11312.

---

> > > ### Author Response · Authors · 2024-11-22
> > > **Reply to follow-up question by Reviewer su1u**
> > >
> > > Thank you for reading our response and for your follow-up question. Indeed, as correctly pointed out by the reviewer, chaotic dynamics inevitably lead to gradient explosion, as shown in the two papers references by the reviewer. This is perfectly in accordance with our previous response clarifying that the Lipschitz constant of the readout MLP would blow up in this case, i.e., can lead to gradient explosions. Note that our underlying linear dynamical system (equation 1) would still be stable, as rigorously shown in the paper. We would like to note here, that this is inevitable to any recurrent model learning chaotic systems, and is not specific to our proposed LinOSS (as it was shown in Hess et al. (2023) and Mikhaeil et al. (2022)). We would further like to note that universality and trainability are two different (but equally important) aspects of our model. While our universality results still holds in the case of chaotic systems, it implies a blow-up of the Lipschitz constant of the readout MLP and thus enables exploding gradients.
> > >
> > > We sincerely hope that we have addressed the concern and question of the reviewer satisfactorily with this response and will add a remark on this to the appendix in a revised version of the paper.

---

> > > > ### Comment · Reviewer_su1u · 2024-11-25
> > > >
> > > > Thank you to the authors for their thoughtful engagement during the rebuttal process. I maintain my positive score.

---

> > > > > ### Author Response · Authors · 2024-11-26
> > > > > **Thanking the Reviewer**
> > > > >
> > > > > We thank the reviewer for your positive assessment of our paper and for appreciating our rebuttal.

---

### Official Review · Reviewer_o2RA · 2024-11-08

**Soundness:** 3
**Presentation:** 4
**Contribution:** 3
**Rating:** 8
**Confidence:** 5

**Summary:**

This paper introduces a new continuous time recurrent network in the family of state space models. The architecture is proposed as an ODE that is discretized in two ways using first order implicit and implicit-explicit integrators. The terms of the ODE are further constrained to induce two different computational tricks to speed up computation; fast matrix inversion to make the implicit methods tractable, and parallel scans for faster sequence processing. The LinOSS architecture is empirically compared to other Neural ODEs and state-space models, and to transformers, on three different time domain problems for time series classification and prediction.

**Strengths:**

The paper demonstrates a cross cutting expertise from dynamical systems analysis through implementation optimization. The design of the ODE is crafting three advantages at three different levels of abstraction simultaneously: enforce theoretically proven stabilization, allow for efficient matrix inversion, and allow for parallel scans of the sequence recurrence. The experiments on time series problems are a good set of problems, and the results of LinOSS stand out against the broad set of comparisons. The proofs are sound, however, it is not clear if they apply to what actually is going on: see below.

**Weaknesses:**

One issue with the paper is highlighted in the core claim of pre hoc controlling for “forgetting” versus stability by choosing between LinOSS-IM and IMEX (Line 298).  Line 205 claims to demonstrate different advantages between the two methods, but this is not actually evident in the experiments. What characteristics of the problems in Table 1 lead to IM vs. IMEX performing differently? If anything, there is no difference between IM and IMEX in all examples but Worms. Is there something special about Worms, or is it a fluke? The result of Table 2 seems to contradict the claim that IMEX is better at long ranges and IM is better at forgetting: why does IM perform better on a problem where memory over a long sequence should be important?

It is odd to rely on the integrator choice to enforce stability vs. forgetting. Using an ODE framework, it would seem more natural to change the ODE with dissipative terms, for example, instead of changing the integrator. By relying on the integrator choice, it is unclear if those properties would actually hold after training, if the discretization is thought to add new properties that the ODE did not have. Given the unclear results of the experiments, is it possible that results of the theoretical analysis do not apply after the model is trained? Did you try inspecting if the parameters of A and S still hold the assumptions / initializations assumed in Section 3, after training?


Consider the case where the system being learned is actually stable, but the backward euler IM formula is applied. Backward Euler is dissipative even when the dynamical system does not want to dissipate. What happens when you try to learn a model that should be energy conserving using  the LinOSS-IM architecture? The authors could try this by just trying to learn to forecast a simple oscillatory system with LinOSS-IM. I would expect that the model would learn “through” the IM discretization, and converge to an parameterization of an “unstable” ODE that is stable after being discretized by IM. See Krishnapriyan, “Learning continuous models for continuous physics” for a discussion on overfitting on learning through ODE discretizations.

One idea to start to tackle this problem: Try LinOSS-FE “forward euler”. The architecture would be similar and the tricks in parallel scans would still work. This would be an ablation that would illustrate why stabilization of the implicit and imex integrators is important. If there is no performance difference with the Forward euler discretization in the experiments, or if the IM method can forecast a stable system, then perhaps the theoretical results do not represent what actually happens after training.

**Questions:**

- What are the runtimes of the different methods in the experiments? While the accuracy metrics are strong, one of the purported examples of SSMs is the efficiency. What are run times when enabling and disabling different introduced optimizations:
  - Run time when not using the equation for matrix inversion?
  - Run time when not using the parallel scan?
- Line 74: Why is A diagonal? Is it only to induce the matrix inversion trick later?
- Some aspects of the architecture are unclear. How exactly are the LinOSS layers stacked into each other and within the network?
  - In Figure 1, which parts of  the figure are u, y, z?
  - What effect do the nonlinearities have on the stability analysis?
  - How is the nonlinear layer in the figure a part of the model?
- In the experiments in Section 4, what are the exact hyperparameters and model graphs?
- The exploitation of Formula 3 to speed up the matrix is clever. How does this differentiate during training, though? Did you pass this formula through autodiff, or define a special differentiation rule?
- Does the parallel scans apply to only using first-order IM or IMEX integrators? Or does the ODE formula in general allow for the parallel scans with other integrators, such as a simple forward Euler, or a higher order IMEX?
- Table 1: s/UAE/UEA/g.
- Table 1: What does UEA stand for? Define abbreviations and add citations. (Is Walker 2024 supposed to be the citation?)
- Color highlighting in the tables is not colorblind friendly, nor BW printer/ereader friendly. Use symbols instead.
- Equation 7: Why is it important that equation 2 is a Hamiltonian system in this section? If the underlying ODE is indeed Hamiltonian, doesn’t that suggest that the IM discretization is not appropriate?
- Line 222: What are a, b supposed to be? Describe the specific case of operator & tuple shown here.
- Line 280: Why is it possible to assume that A_kk>0? Couldn’t they diverge from that assumption during training? What initialization is required?
- Line 289: s/constraint/constrained/
- Line 292: The steps in the proof are not obvious. More steps of proof should be presented. In the appendix would be sufficient.
- Just a remark: The title of section 4.2 is overly extravagant for the claims. These days, “extreme long ranges” would be context lengths of millions of tokens :)
- Appendix A: What are the parameter counts? What is the architecture of the “nonlinear layers”? What are the hyperparameters for the forecasting problem?
- Why does the PPG model have higher memory demands, when the models actually seem small?
- Appendix A: What ML library was used?
- Could you describe the loss functions and the complete architecture for the 3 different types of problems? How is time series classification grafted onto the LinOSS network?
- The importance of section 3.2 is unclear to me. It is nice to prove universality of a model, but what is special about LinOSS that other more general proofs of universality would not apply? I would not have questioned it. Is any aspect of the proof specific to LinOSS, or could it apply more broadly to more SSMs?

---

> ### Author Response · Authors · 2024-11-21
> **Reply to Reviewer o2RA Part 1**
>
> We start by thanking the reviewer for appreciating the merits of our paper and the welcome suggestions to improve it. Below, we address the very valid concerns raised by the reviewer and thank the reviewer in advance for their patience in reading our detailed reply.
>
> - **Connection between theoretical findings and empirical results:** We apologize for the ambiguous statement in our first version of the paper regarding forgetting in LinOSS-IM vs LinOSS-IMEX. We would like to emphasize that the main message of our paper is that we use forced harmonic oscillators as a building block to construct a new state-space model architecture. As the reviewer correctly pointed out, the 'proper' way to dicretize this ODE system is via symplectic time integration method, e.g., implicit-explicit Euler as in the case of LinOSS-IMEX. We argue, however, that there is another viable choice to discretize this system using purely implicit methods. These methods, in contrast to symplectic methods, introduce dissipation to the system. However, we further argue that this can actually be an advantage when learning long-range sequences, as it introduces 'forgetting' in the sense of the equation in lines 260-262. This makes the model more flexible while at the same time can still propagate information over very long timescales, i.e., the absolute eigenspectrum of the implicit LinOSS can be further controlled via the state matrix (equation line 289) and thus can be made arbitrarily close to 1. This flexibility in balancing forgetting with propagating information over long timescales makes LinOSS-IM a strong candidate for learning on long-range sequences. We would thus expect that LinOSS-IM outperforms LinOSS-IMEX in practice unless the underlying dataset contains conserved quantities. We have clarified this now in Remark 1 in the revised paper.
> We further followed the reviewer's excellent suggestion and provide a new synthetic experiment examining the dissipative nature of LinOSS-IM vs the conservative one of LinOSS-IMEX. To this end, we predict simple harmonic motion for different initial positions and velocities for a long time. We show that LinOSS-IMEX keeps the error constant, while the error of LinOSS-IM grows over time. This is due to the fact that LinOSS-IM introduces dissipative behavior and in the asymptotic limit would converge towards a steady-sate for any state matrix values. This is in contrast to the conservative nature of LinOSS-IMEX. We thank the reviewer again for suggesting this experiment. We have added the experiment to the paper in Section C.2 and a discussion about it in Section 4.4.
> Finally, we would like to point out that we have implemented an explicit version of LinOSS using the explicit Euler method. However, having tested this architecture on several datasets it continues to return NaN values after some time, since the hidden states of this system explode. Of course, one could take the limit of $\Delta t\rightarrow0$ to stabilize it, but this would be in contrast to the idea of efficiency in state-space models.
>
> - **"What are the runtimes of the different methods in the experiments? Run time when not using the equation for matrix inversion? Run time when not using the parallel scan?":** We have now added run times, GPU memory usage, and number of parameters for all models on all six datasets from Section 4.1 in Table 5 in the Appendix. While LinOSS seems to be well within the efficiency range of other state-space models, we would like to highlight that LinOSS is the fastest of all other considered models on two out of six datasets, as well as the second fastest on again two out of six datasets. We further note that relying on numerical methods to invert the state matrix would make associative parallel scans obsolete, as the run time is proportional to $log_2(N)$ up to some constant that depends on the structure of the state matrix. Moreover, not using a parallel scan would result in a simple linear RNN cell stacked over several layers. This is known to be exceptionally slow.
>
> - **"Why is A diagonal":** The majority of recent state-space models choose A to be diagonal. Indeed, this allows for a quick closed-form matrix inversion in our case. Moreover, the resulting matrix $\bf M$ (eq 4 and eq 6) is a 2 x 2 block matrix, where each block is again a diagonal matrix. This leads to a very fast computation of the ODE solution using associative scans. Note that associative parallel scans are significantly slower for dense state matrices. Finally, it was noted in Orvieto et al, 2023, (see Table 2 of this paper) that diagonal state matrices perform better in practice compared to dense matrices.

---

> > ### Author Response · Authors · 2024-11-21
> > **Reply to Reviewer o2RA Part 2**
> >
> > - **''Some aspects of the architecture are unclear'':** We apologize for not having provided all necessary architectural details in the previous version of the paper. We have now added a new section in the appendix (Section A), which provides the details of the full LinOSS architecture, including the nonlinear layers within a LinOSS block. We further provide the full LinOSS model using pseudo-code in Algorithm 1 in the Appendix. Moreover, we added details to Figure 1, labeling $\bf u(t)$ and $\bf y(t)$ in the figure.
> >
> > - '**'What effect do the nonlinearities have on the stability analysis?''** The nonlinear layers are applied element-wise over time. They are not part of the recurrence and thus have no influence on the recurrent stability of the model. Therefore, any nonlinear block (e.g., MLP) can be used in this context.
> >
> > - **''In the experiments in Section 4, what are the exact hyperparameters and model graphs?''** We have provided all hyperparameters in Section B.1 of the Appendix. Moreover, we provide the code to reproduce our results.
> >
> > - **''The exploitation of Formula 3 to speed up the matrix is clever. How does this differentiate during training, though? Did you pass this formula through autodiff, or define a special differentiation rule?''** We are glad the reviewer appreciates our contribution in this context. Indeed, we use equations 4 and 6 for the autodifferentiation.
> >
> > - **''Does the parallel scans apply to only using first-order IM or IMEX integrators?''** This is a very interesting question. Motivated by this, we have added a new section in the Appendix (Section D), outlining the process of leveraging higher-order time integration schemes in this context. Moreover, we provide an explicit example of using the symplectic second-order velocity Verlet method for LinOSS. We note, however, that higher-order systems have a higher computational cost and would thus decrease the efficiency of LinOSS. Moreover, since we are not trying to solve an ODE as accurate as possible, but simple want to use some structure of the underlying ODE system to build a new model architecture, we conclude that fast first-order methods are sufficient in our context. However, we would be interested in studying the effects of higher-order methods on LinOSS in detail in future work.
> >
> > - **''Equation 7: Why is it important that equation 2 is a Hamiltonian system in this section? If the underlying ODE is indeed Hamiltonian, doesn’t that suggest that the IM discretization is not appropriate?''** We mention this to highlight the fact that LinOSS has a corresponding Hamiltonian. Indeed, the correct way to solve Hamiltonian systems is to use symplectic methods. Our goal, however, was to show that using another discretization, i.e., fully implicit discretization, leads to a model with other favorable properties, which is potentially even more flexible compared to using a symplectic method.
> >
> > - **''Why is it possible to assume that $A_{kk}\geq0$?''** We parameterize our state matrix in such a way that this is always satisfied, using either ReLU activation functions or squaring the state matrix diagonal elements. We provide a discussion on this in Section 3.1. under paragraph "Initialization and parameterization of weights.". Moreover, we empirically test these parameterizations in Section C.1 in the Appendix.
> >
> > - **''Line 292: The steps in the proof are not obvious. More steps of proof should be presented.''** We apologize for not having provided more details about the proof in the first version of this paper. We fully agree with the reviewer on that and provide the proposition with the proof in Section E.1. in the Appendix of the revised version.
> >
> > - **''Why does the PPG model have higher memory demands, when the models actually seem small?''** Most of the memory requirements stem from the underlying sequences, i.e., the memory requirement scales with the length of the sequences. Since PPG is much longer than any of the other considered datasets it requires the most GPU memory.
> >
> > - **''Appendix A: What ML library was used?''** We have used JAX for all our experiments. We have clarified this now at the beginning of Section B.

---

> > > ### Author Response · Authors · 2024-11-21
> > > **Reply to Reviewer o2RA Part 3**
> > >
> > > - **''Could you describe the loss functions and the complete architecture for the 3 different types of problems? How is time series classification grafted onto the LinOSS network?''** We have added a new section in the appendix (Section A.2) describing the procedure of leveraging LinOSS (and any other state-space model for that matter) for sequence classification, regression, and time-series forecasting.
> > >
> > > - **''The importance of section 3.2 is unclear to me. It is nice to prove universality of a model, but what is special about LinOSS that other more general proofs of universality would not apply? I would not have questioned it. Is any aspect of the proof specific to LinOSS, or could it apply more broadly to more SSMs?''** Our universality proof is indeed specific to LinOSS, as its main statement reads that any such operator between infinite dimensional spaces can be encoded by a finite dimensional one using time-windowed sine transforms, i.e., solutions of forced harmonic oscillators (or LinOSS). We further argue that the universality of these models (including also other oscillatory methods) is actually surprising, in fact, this was not proven until just a few months ago (see Lanthaler et al, NeurIPS 2023).
> > >
> > > We have further changed and corrected any other minor comment raised by the reviewer.
> > >
> > > We sincerely hope that we have addressed the concerns of the reviewer, particularly on the connection between the theoretical part and empirical results, satisfactorily in the revised version and would kindly ask the reviewer to update their score accordingly.

---

> > > > ### Comment · Reviewer_o2RA · 2024-11-26
> > > >
> > > > I thank the author’s for the detailed response and changes to the paper. The revisions and discussions address all my concerns. I have thus raised my score. Detailed responses below:
> > > >
> > > > The Forward Euler result fits in with theoretical expectations, though I am surprised as it often performs better than expected; NeuralODEs with forward euler can overfit to a stable model. I recommend mentioning that the model NaNs very quickly during training in the paper, since it is an empirical validation that the complication of IM and IMEX integrations are necessary. A one line statement would be sufficient.
> > > >
> > > > Figure 1 is still a little unclear: y and z are the internal state of the ode model, which is very important but where are they in the figure? I understood that there are 2 continuous trajectories that are integrated (y and z) and then there are a continuous input signal and output signal that are not integrated.

---

> > > > > ### Author Response · Authors · 2024-11-26
> > > > > **Thanking the Reviewer**
> > > > >
> > > > > We thank the reviewer for appreciating our rebuttal and revised version and for raising their score.
> > > > >
> > > > > We have revised our manuscript and have added a comment about explicit time integration methods such as explicit Euler at the end of Remark 1. We further agree with the reviewer that showing ${\bf y}(t)$ and ${\bf x}(t)$ is more intuitive than showing ${\bf x}(t)$ in Figure 1. We have updated the Figure and the caption accordingly.

---

### Author Response · Authors · 2024-11-21
**Reply to all the reviewers**

We would like to thank all four reviewers for their thorough and patient reading of our article and for praising our paper as, ''demonstrates a cross cutting expertise from dynamical systems analysis through implementation optimization'' (o2RA), ''provides strong theoretical together with intuitive explanations'' (su1u), ''demonstrates strong theoretical foundations by providing rigorous mathematical analysis'' (Wn4t), and that our ''work is well motivated'' (w9Vn). We are also glad the reviewers appreciated our experimental results and described them as ''excellent and definitely contributes to the field significantly.'' (su1u), ''stand out against the broad set of comparisons'' (o2RA), ''great, showing strong performance on diverse tasks'' (Wn4t), and ''thorough and demonstrate the efficacy of the proposed approach'' (w9Vn).

The constructive suggestions allowed us improve the paper, which we hope the reviewers will find to satisfactorily address the raised concerns.

We uploaded a revised version of the paper incorporating the reviewers' suggestions. All changes are highlighted in blue color.

The following main changes were made:

- Remark 1: clarifying ambiguities regarding dissipation vs conservation of LinOSS-IM and LinOSS-IMEX
- Section 4.4: added new experimental results regarding sensitivity of LinOSS performance to the timestep $\Delta t$, different state matrix initializations, and different state matrix parameterizations.
- Appendix Section A: added details about full LinOSS architecture further explaining Figure 1. Moreover, clarifying sequence-to-sequence architecture for time-series forecasting tasks.
- Appendix Table 5: added run time, GPU memory usage, and number of parameters for all models and tasks considered in Section 4.1.
- Appendix Section C.2: added new experiment addressing the role of dissipation vs conservation in LinOSS.
- Appendix Section D: added a section about leveraging higher-order time integration schemes, as well as providing a full example of using second-order velocity Verlet in this context.
- Appendix Section E.1: added detailed proposition and proof about eigenspectrum of LinOSS-IMEX.

We answer the points raised by each of the reviewers individually in our comments below. Please note that all the references to page numbers, sections, figures, tables, equation numbers and references, refer to those in the revised version. We hope that based on the answers to their questions and the improvements made to the revised version, the reviewers will increase their scores.

---

### Meta-Review · Area_Chair_Utz5 · 2024-12-18

**Metareview:**

It is a pleasure when the reviewers all converge on a consistent recommendation. I see no reason to disagree with the universal praise for the paper, and strongly recommend acceptance, and believe the paper should be highlighted with a spotlight presentation.

**Additional Comments On Reviewer Discussion:**

Most of the reviewers were detailed and engaged in the discussion.

---

### Decision · Program_Chairs · 2025-01-22

Accept (Oral)